# An improved fluorescent noncanonical amino acid for measuring conformational distributions using time-resolved transition metal ion FRET

William N Zagotta[1]*, Brandon S Sim[1], Anthony K Nhim[1], Marium M Raza[1], Eric GB Evans[1], Yarra Venkatesh[2], Chloe M Jones[2,3], Ryan A Mehl[4], E James Petersson[2], Sharona E Gordon[1]*

[1]Department of Physiology and Biophysics, University of Washington, Seattle, United States; [2]Department of Chemistry, University of Pennsylvania, Philadelphia, United States; [3]Biochemistry and Molecular Biophysics Graduate Group, University of Pennsylvania, Philadelphia, United States; [4]Department of Biochemistry and Biophysics, Oregon State University, Corvallis, United States

**Abstract** With the recent explosion in high-resolution protein structures, one of the next frontiers in biology is elucidating the mechanisms by which conformational rearrangements in proteins are regulated to meet the needs of cells under changing conditions. Rigorously measuring protein energetics and dynamics requires the development of new methods that can resolve structural heterogeneity and conformational distributions. We have previously developed steady-state transition metal ion fluorescence resonance energy transfer (tmFRET) approaches using a fluorescent noncanonical amino acid donor (Anap) and transition metal ion acceptor to probe conformational rearrangements in soluble and membrane proteins. Here, we show that the fluorescent noncanonical amino acid Acd has superior photophysical properties that extend its utility as a donor for tmFRET. Using maltose-binding protein (MBP) expressed in mammalian cells as a model system, we show that Acd is comparable to Anap in steady-state tmFRET experiments and that its long, single-exponential lifetime is better suited for probing conformational distributions using time-resolved FRET. These experiments reveal differences in heterogeneity in the apo and holo conformational states of MBP and produce accurate quantification of the distributions among apo and holo conformational states at subsaturating maltose concentrations. Our new approach using Acd for time-resolved tmFRET sets the stage for measuring the energetics of conformational rearrangements in soluble and membrane proteins in near-native conditions.

*For correspondence:
zagotta@uw.edu (WNZ);
seg@uw.edu (SEG)

Competing interest: The authors declare that no competing interests exist.

## Introduction

Proteins are exquisite molecular machines that underlie virtually all physiological functions. Proteins act as sensors for signaling molecules, catalyze chemical reactions, change membrane electrical potential, and generate mechanical force, all by changing their distribution among conformational states. Whereas structural methods like cryo-electron microscopy (cryo-EM) and X-ray crystallography have provided high-resolution information on the architecture of some of these different conformations, they provide little information about the structural dynamics: the rates and energetics associated with the conformational transitions (*Cheng, 2018*). These energetics are the basis of protein function and understanding them promises to be one of the next frontiers in biology.

Most current biochemical methods provide either structural or energetic information, but not both. To elucidate protein dynamics, a method should ideally (1) measure dynamics with high temporal resolution, over a large range of time scales (from nanoseconds to minutes); (2) measure structure with high spatial resolution; (3) resolve heterogeneity and distributions of conformations; (4) exhibit high sensitivity, allowing measurements from small amounts of protein, even single molecules; (5) work on proteins of arbitrary size; (6) work on proteins in their native environments, including membranes and protein complexes; (7) be minimally perturbing; and (8) measure structural dynamics and function simultaneously. While many methods, such as cryo-EM, nuclear magnetic resonance (NMR), and electron paramagnetic resonance (EPR), satisfy some of these criteria, no current method satisfies all of them.

One approach that has the potential to satisfy all of these criteria is fluorescence resonance energy transfer (FRET). FRET is the nonradiative, through-space transfer of energy from a donor fluorophore to a nearby acceptor chromophore (*Stryer and Haugland, 1967*; *Lakowicz, 2006*). The FRET efficiency is steeply dependent on distance between the donor and acceptor, $r$, decreasing as $\frac{1}{r^6}$, making FRET a precise molecular ruler for measurements of distances between these probes. Each donor-acceptor pair has a characteristic distance, called the Förster distance ($R_0$), at which FRET is 50% efficient. FRET measurements can be made from any size protein in vitro or in its native cellular environment. Fluorescence measurements are highly sensitive, giving FRET single-molecule sensitivity and high temporal resolution to record submillisecond events. Although FRET can only measure one set of distances in the protein at a time, the distances can have high spatial resolution (<2 Å) and can be interpreted in the framework of high-resolution static structures obtained from other methods (*Gordon et al., 2018*).

One particularly powerful approach is to measure the FRET efficiency using fluorescent lifetimes, also known as time-resolved FRET (*Grinvald et al., 1972*). The lifetime of a fluorophore is the time between its excitation and emission of a photon, typically in the nanosecond range (*Lakowicz, 2006*). Steady-state FRET measurements record the intensity of emission using constant illumination and report a weighted-average distance between probes. In contrast, time-resolved FRET uses either pulsed or intensity-modulated excitation light and records the emission using a high-speed detection system. Time-resolved FRET can reveal the distribution of the distances between probes, something not possible using standard steady-state FRET approaches (*Grinvald et al., 1972*; *Haas et al., 1975*; *Lakowicz et al., 1987b*). Thus, time-resolved FRET can provide a nanosecond snapshot of the protein that reveals the conformational heterogeneity and the energetics of the conformational rearrangement.

FRET measurements typically measure distances in the 30–80 Å range, limiting their utility for measuring intramolecular distances and distance changes associated with conformational changes. To overcome these limitations, we have previously developed a method called transition metal ion FRET (tmFRET) to accurately measure the structure and dynamics of short-range interactions in proteins (*Taraska et al., 2009a*; *Taraska et al., 2009b*; *Yu et al., 2013*; *Dai et al., 2018*). tmFRET utilizes FRET between a fluorophore and a transition metal divalent cation to measure intramolecular distances. Transition metal cations, such as $Ni^{2+}$, $Co^{2+}$, and $Cu^{2+}$, act as nonfluorescent acceptors, quenching the fluorescence of donor fluorophores with an efficiency that is steeply distance dependent over short distances (10–25 Å) (*Latt et al., 1970*; *Latt et al., 1972*; *Horrocks et al., 1975*; *Richmond et al., 2000*; *Sandtner et al., 2007*). The donor fluorophore can be introduced at one site in the protein either by labeling the protein with a cysteine-reactive fluorophore (*Taraska et al., 2009a*; *Yu et al., 2013*) or as a fluorescent noncanonical amino acid (*Kalstrup and Blunck, 2013*; *Zagotta et al., 2016*; *Dai and Zagotta, 2017*; *Dai et al., 2018*; *Gordon et al., 2018*; *Dai et al., 2019*; *Dai et al., 2021*). The transition metal acceptor-binding site can be introduced either as a cysteine-reactive metal ion chelator (*Taraska et al., 2009b*; *Gordon et al., 2018*) or as a di-histidine motif (*Taraska et al., 2009a*). In each case, the small probe is attached to the protein with a minimal linker, making tmFRET a more faithful reporter of intramolecular rearrangements than traditional FRET, which often involves long, flexible linkers to large fluorophores such as fluorescein and rhodamine derivatives.

Recently, we have introduced a variation of tmFRET called ACCuRET (Anap Cyclen-$Cu^{2+}$ resonance energy transfer) (*Gordon et al., 2018*). ACCuRET uses the fluorescent noncanonical amino acid L-3-(6-acetylnaphthalen-2-ylamino)–2-aminopropanoic acid (Anap) as the FRET donor. Anap is incorporated into the protein using amber codon suppression methods and an engineered aminoacyl

tRNA synthetase (RS) that selectively charges an orthogonal tRNA with Anap (***Chatterjee et al., 2013***; ***Kalstrup and Blunck, 2013***). The FRET acceptor is $Cu^{2+}$ bound to TETAC (1-(2-pyridin-2-yldisulfanyl) ethyl)–1,4,7,10-tetraazacyclododecane, a cysteine-reactive cyclen molecule that chelates $Cu^{2+}$ with subnanomolar affinity (***Kodama and Kimura, 1977***). The FRET efficiency is measured by the decrease in Anap fluorescence after labeling of an engineered cysteine residue by $Cu^{2+}$-TETAC. Using maltose-binding protein (MBP) as a benchmark, we showed that tmFRET accurately measures short-range distances and small changes in distance (***Gordon et al., 2018***). This method has subsequently been used to measure the conformational rearrangements in various ion channels including KCNH channels and HCN channels (***Dai et al., 2018***; ***Dai et al., 2019***; ***Dai et al., 2021***).

In this study, we extended our tmFRET approach by introducing a fluorescent donor that is better suited for measuring distance distributions using time-resolved tmFRET. L-Acridonylalanine (Acd) is a small fluorescent noncanonical amino acid that shows great promise as a tmFRET donor (***Speight et al., 2013***). Acd has previously been incorporated into *Escherichia coli* proteins using an engineered RS based on a *Methanocaldococcus jannachii* (Mj) tyrosyl RS that is orthogonal only to the translation machinery in prokaryotes (***Speight et al., 2013***; ***Padmanarayana et al., 2014***; ***Sungwienwong et al., 2017***; ***Hostetler et al., 2018***; ***Hostetler et al., 2020***; ***Jones et al., 2020***). Recently, we engineered a pyrrolysine (Pyl) RS from *Methanosarcina barkeri* (Mb) that, together with its tRNA, can specifically incorporate Acd in both prokaryotic and eukaryotic expression systems (***Jones et al., 2021***). Here we compare Acd to Anap as a donor for tmFRET. We show that Acd overcomes some of the limitations of Anap and is in many ways an improvement over our previous ACCuRET methods. Furthermore, we show that Acd can be used in time-resolved FRET experiments to measure structural heterogeneity within each conformational state of MBP and the distribution of MBP among different conformational states. This capability can be used, in principle, to measure the energetics of any conformational rearrangement of a protein domain slower than tens of nanoseconds.

## Results

### Acd vs. Anap

Previously, we used the fluorescent ncAA Anap for tmFRET experiments. However, Acd has a number of fluorescence properties that make it better suited for tmFRET than Anap. With the new Acd RS derived from Pyl-RS, Acd can be used for both prokaryotic and eukaryotic expression (***Jones et al., 2021***). Acd is similar in size to Anap, with three rings instead of two but with a linker to the α-carbon that is shorter by one rotatable bond; therefore, Acd has fewer possible rotameric states (***Figure 1A,B***). Like Anap, Acd absorbs in the far UV, but with a peak absorption at 385 nm instead of 350 nm for Anap. This makes Acd more compatible with the optics of most light microscopes and 405 nm laser excitation, allowing for greater excitation with lower autofluorescence from the cell. Anap and Acd both emit in the visible region with spectra that overlap the absorption of $Cu^{2+}$-TETAC (***Figure 1C,D***) and other transition metals, making them both compatible with tmFRET. Acd, however, is much less environmentally sensitive than Anap. The emission of Anap is brighter and blue-shifted in a hydrophobic environment, whereas the emission of Acd is less affected by these solvents (compare DMSO and EtOH to KBT, our standard intracellular buffer, in ***Figure 1C,D***), alleviating concerns that environmental changes in quantum yield will affect $R_0$. In an aqueous environment, Acd has a lower extinction coefficient (7300 $M^{-1}$ $cm^{-1}$ at 385 nm for Acd vs. 12,600 $M^{-1}$ $cm^{-1}$ at 350 nm for Anap), but a larger quantum yield (0.8 for Acd vs. 0.3 for Anap), so that, with the same excitation light intensity, Acd is modestly brighter than Anap in an aqueous environment (but dimmer in a hydrophobic environment; ***Figure 1A, B and G***). Acd also photobleaches about 10 times slower than Anap (***Figure 1H***), allowing for greater intensity excitation, brighter fluorescence, and experiments of longer duration. In addition, Acd incorporated into proteins is robust in the cell environment, whereas Anap appears to undergo a chemical change inside the cell at some protein sites, dramatically decreasing its brightness and blue-shifting its emission spectrum (***Figure 1—figure supplement 1***).

Another property of Acd that makes it superior to Anap and many other fluorophores is its fluorescence lifetime. We measured the fluorescence lifetime of free Acd and Anap using a frequency-domain lifetime instrument (see Materials and methods). For frequency-domain measurements, the lifetime is described by plots of the shift of the phase of the response (phase delay) and the decrease in the amplitude of the response (modulation ratio) as a function of the modulation frequency of the

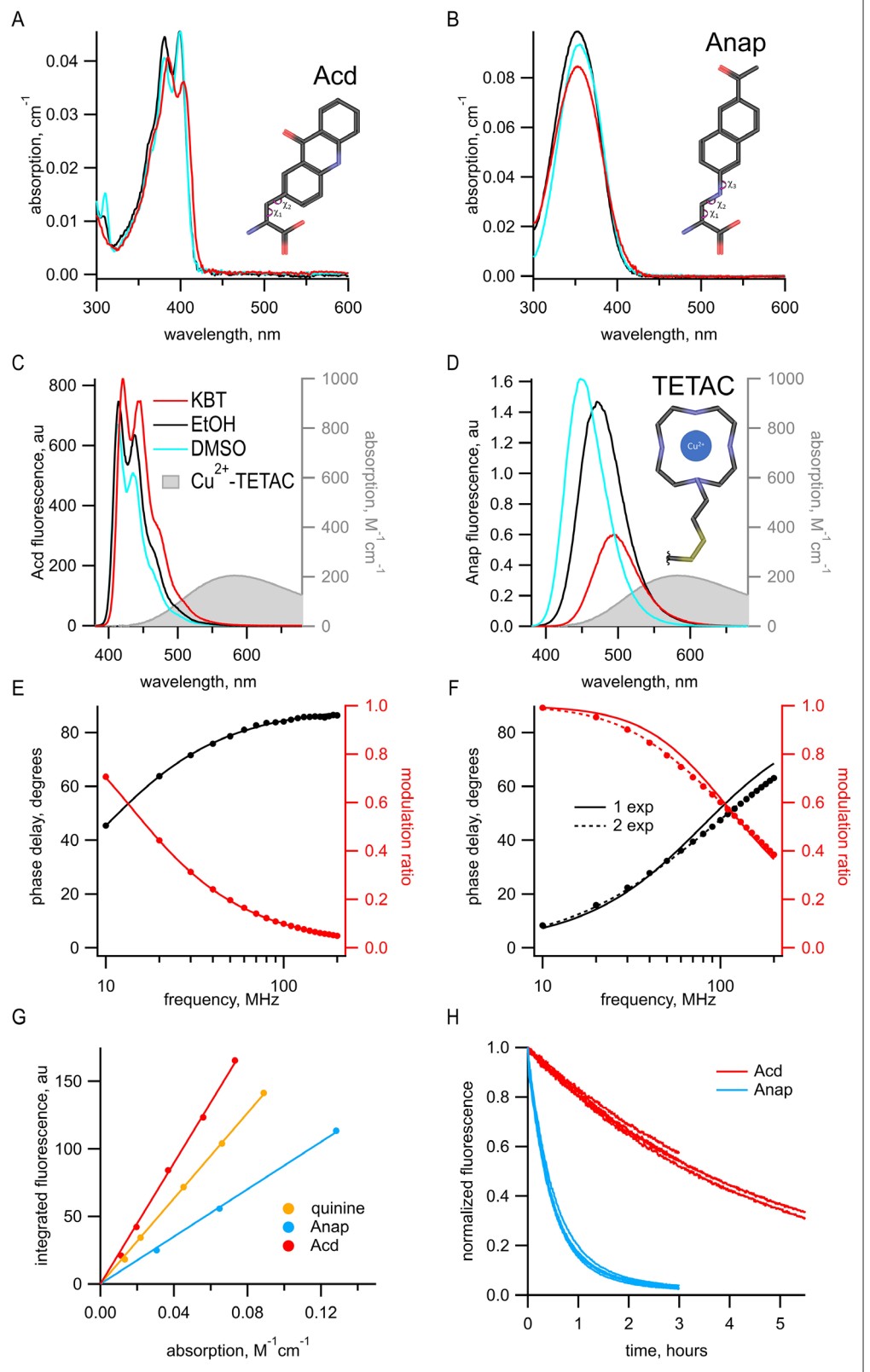

**Figure 1.** Acd has improved photophysical properties compared to Anap. (**A, B**) Absorption spectra of (**A**) Acd and Anap (**B**) in aqueous solution (KBT; red), EtOH (black), and DMSO (cyan), with the structures of the amino acids, inset. (**C, D**). Emission spectra in response to 370 nm excitation (left axis) and absorption spectrum of Cu²⁺-TETAC (right axis) in KBT for Acd (**C**) and Anap (**D**). Color scheme is the same as (**A**), but with Cu²⁺-TETAC

*Figure 1 continued on next page*

*Figure 1 continued*

spectrum in gray. (**E, F**) Frequency-domain measurements of fluorescence lifetime of Acd (**E**) and Anap (**F**) measured in KBT. Black: phase delay in degrees; red: modulation ratio. Curves are fits with **Equations 5** and **6** corrected by **Equations 13** and **14**. Fit parameters for (**E**) are $\tau_{D_1}$ = 16.1 ns and for (**F**) are $\tau_{D_1}$ = 2.0 ns for a single exponential (solid curve) and $\tau_{D_1}$ = 1.3 ns, $\tau_{D_2}$ = 3.3 ns, and $\alpha_1$ = 0.76 (dashed curve). See Table 1 for collected data. (**G**) Quantum yield for Acd (red) and Anap (blue) was determined relative to quinine in 0.5 M $H_2SO_4$ (gold), as described in Materials and methods. Solid lines are fits with a line through the origin. (**H**) Acd (red) is more resistant to photobleaching than Anap (blue). For Acd, excitation at 380 nm with 14.7 nm slits and emission at 450 nm with 1 nm slit. For Anap, excitation at 350 nm with 14.7 nm slits and emission at 490 nm with 1 nm slit. Note that the Acd excitation light was ~38% brighter than the Anap excitation light. For both Acd and Anap, four independent experiments were each fit with a single exponential. The mean time constants ± SEM were 18,000 ± 863 s (n = 4) for Acd and 1838 ± 52 s (n = 4) for Anap.

The online version of this article includes the following figure supplement(s) for figure 1:

**Figure supplement 1.** Anap, but not Acd, appeared to undergo a site-specific chemical change in cells.

excitation light. These data can then be fit with models for the lifetimes that assume that they have single-exponential or multiexponential decays. The data for Anap were best fit by two exponentials with time constants $\tau_{D_1}$ = 1.3 ns and $\tau_{D_2}$ = 3.3 ns, and with the relative amplitude $\alpha_1$ = 0.76 (**Figure 1F** and **Table 1**). In contrast, the measured lifetime for Acd was single exponential with a $\tau_{D_1}$ = 16 ns

**Table 1.** Fit parameters for frequency-domain lifetime data.

| Donor | Acceptor | Recovery | | $\tau_{D1}$ (ns) | $\alpha_1$ | $\tau_{D2}$ (ns) | $\bar{r}_{D1}$(Å) | $\sigma_1$ (Å) | $\bar{r}_{D2}$(Å) | $\sigma_2$ (Å) | $A_2$ 0.2 mM | $A_2$ 0.37 mM |
|---|---|---|---|---|---|---|---|---|---|---|---|---|
| Anap | n/a | | | | | | | | | | | |
| | | Mean | | 1.3 | 0.76 | 3.3 | | | | | | |
| | | SEM | | 0.02 | 0.003 | 0.02 | | | | | | |
| | | n | | 4 | 4 | 4 | | | | | | |
| Acd | n/a | | | | | | | | | | | |
| | | Mean | | 16.0 | | | | | | | | |
| | | SEM | | 0.03 | | | | | | | | |
| | | n | | 4 | | | | | | | | |
| MBP-295Acd-Y307F-C | Cu²⁺-TETAC | | | | | | | | | | | |
| | | Mean | 0.92 | 15.6 | | | 23.7 | 6.2 | 13.4 | 6.5 | | |
| | | SEM | 0.040 | 0.05 | | | 0.36 | 0.47 | 0.38 | 0.27 | | |
| | | n | 10 | 10 | | | 10 | 10 | 7 | 7 | | |
| MBP-322Acd-C | Cu²⁺-TETAC | | | | | | | | | | | |
| | | Mean | 0.94 | 15.1 | | | 13.6 | 5.3 | 15.8 | 5.5 | | |
| | | SEM | 0.015 | 0.09 | | | 0.06 | 0.22 | 0.09 | 0.15 | | |
| | | n | 5 | 5 | | | 5 | 5 | 4 | 4 | | |
| MBP-295Acd-Y307F-HH | Cu²⁺ | | | | | | | | | | | |
| | | Mean | 0.88 | 15.5 | | | 18.3 | 2.5 | 12.7 | 1.3 | 0.38 | 0.60 |
| | | SEM | 0.040 | 0.07 | | | 0.21 | 0.54 | 0.04 | 0.26 | 0.002 | 0.02 |
| | | n | 5 | 5 | | | 3 | 3 | 4 | 4 | 3 | 3 |
| MBP-295Acd | n/a | | | | | | | | | | | |
| | | Mean | | 14.0 | 0.70 | 2.0 | | | | | | |
| | | SEM | | 0.14 | 0.011 | 0.08 | | | | | | |
| | | n | | 5 | 5 | 5 | | | | | | |

$\bar{r}_{D1}$ and $\sigma_1$ are from fits to the data in the absence of maltose. $\bar{r}_{D2}$ and $\sigma_2$ are from fits to the data in the presence of 10 mM maltose. Recovery refers to fits to the data in the presence of TCEP, for Cu²⁺-TETAC, and EDTA, for Cu²⁺.

(*Figure 1E* and *Table 1*). The longer lifetime of Acd provides a greater dynamic range for FRET and fluorescence polarization measurements (*Hostetler et al., 2020*), and the single-exponential distribution greatly simplifies the analysis of FRET (see below).

## Acd incorporation into MBP

To investigate the utility of Acd as a tmFRET donor, we incorporated Acd into MBP in mammalian cells. MBP is a clamshell-shaped protein that undergoes a significant closure of the clamshell upon binding maltose (see Figure 3A,B). For these experiments, we used two donor sites for specific incorporation of Acd, amino acid 295 at the outer lip of the clamshell and 322 on the back side of the clamshell. We have previously shown that Anap incorporated at MBP-295 and MBP-322 can be paired with nearby acceptor sites to measure maltose-dependent decreases or increases in distance, respectively (*Gordon et al., 2018*). Our MBP constructs have a mutation in the maltose-binding site, W340A, previously shown to decrease the affinity of MBP for maltose to ~300 µM (*Martineau al., 1990*) and prevent the binding of endogenous sugars during protein purification from mammalian cells (*Gordon et al., 2018*). Using MBP allowed us to test if tmFRET could measure distances, distance distributions, and state energetics in a protein with a well-characterized structure and conformational rearrangement.

To site-specifically incorporate Acd into MBP, we used amber codon suppression (*Nikić and Lemke, 2015*). We have recently engineered a Pyl-RS (Acd-RS) that, together with its orthogonal tRNA, specifically incorporates Acd in both prokaryotic and eukaryotic expression systems (*Jones et al., 2021*). For this study, our amber codon suppression method involved co-transfecting mammalian HEK293T/17 cells with a plasmid encoding the Acd-RS/tRNA pair and a second plasmid encoding MBP with a TAG stop codon (the amber codon) engineered at the site for Acd incorporation. Acd was then added to the culture medium, and the cells were harvested after 1–2 days.

Acd incorporation into MBP at sites 295 and 322 was highly specific. In-gel fluorescence revealed that full-length MBP-Acd was the primary band in clarified cell lysates and was not present without co-transfecting the Acd-RS/tRNA plasmid (*Figure 2A,B*). The MBP was affinity purified to homogeneity using its amino-terminal FLAG tag (*Figure 2C,D*). Anti-FLAG western blots revealed that, whereas it was entirely truncated at the TAG sites without Acd-RS, the majority of the MBP was full length with Acd-RS (*Figure 2E,F*). The fraction of MBP that was truncated with Acd-RS was substantially reduced by co-transfection with a plasmid encoding a dominant negative eRF1 (eRF1-E55D; data not shown) that was previously reported to decrease termination at TAG sites during amber codon suppression with these MBP constructs (*Schmied et al., 2014*; *Gordon et al., 2018*).

Fluorescence-detection size-exclusion chromatography (FSEC) revealed a single monodispersed peak for purified MBP-295Acd and MBP-322Acd (excitation: 385 nm; emission: 450 nm) but only a minor signal without Acd-RS (*Figure 2G*). The UV absorption (280 nm) displayed two primary peaks, one for MBP and one for the FLAG peptide used to elute MBP from the anti-FLAG beads (*Figure 2H*). Interestingly, whereas the UV absorption of MBP-295Acd and MBP-322Acd was very similar in amplitude, the fluorescence of MBP-295Acd was only about half that of MBP-322Acd (*Figure 2G,H*). As shown below, the lower specific fluorescence of MBP-295Acd compared to MBP-322Acd was largely due to the partial quenching of Acd by a nearby tyrosine (Y307) in MBP-295Acd.

## tmFRET with $Cu^{2+}$-TETAC for MBP-295Acd and MBP-322Acd

To determine if Acd could serve as a donor for tmFRET, we introduced single cysteine mutations into our MBP-Acd constructs for modification by the cysteine-reactive acceptor $Cu^{2+}$-TETAC. For MBP-295Anap, cysteine was introduced at position 237 (MBP-295Acd-C) and, for MBP-322Acd, cysteine was introduced at position 309 (MBP-322Acd-C) (*Figure 3A,B*). These donor-acceptor pairs were selected because (1) the sites are solvent exposed, (2) the sites are on rigid secondary structural elements (α-helices in MBP), (3) the distances between the sites are predicted to fall in the working range of tmFRET (~10–25 Å), and (4) the distances between the sites undergo moderate changes between apo and holo MBP. *Figure 3A,B* shows the position of these donor (magenta) and acceptor (orange) sites as clouds of possible rotamers modeled onto the X-ray crystal structures of the apo and holo states of MBP. MBP-295Acd-C has donor and acceptor sites on the outer lip of the clamshell with β-carbon distances, measured from the X-ray structures, of 21.3 Å in the apo state and 12.9 Å in the holo state, a distance change of −8.4 Å (shortening). MBP-322Acd-C has donor and acceptor sites

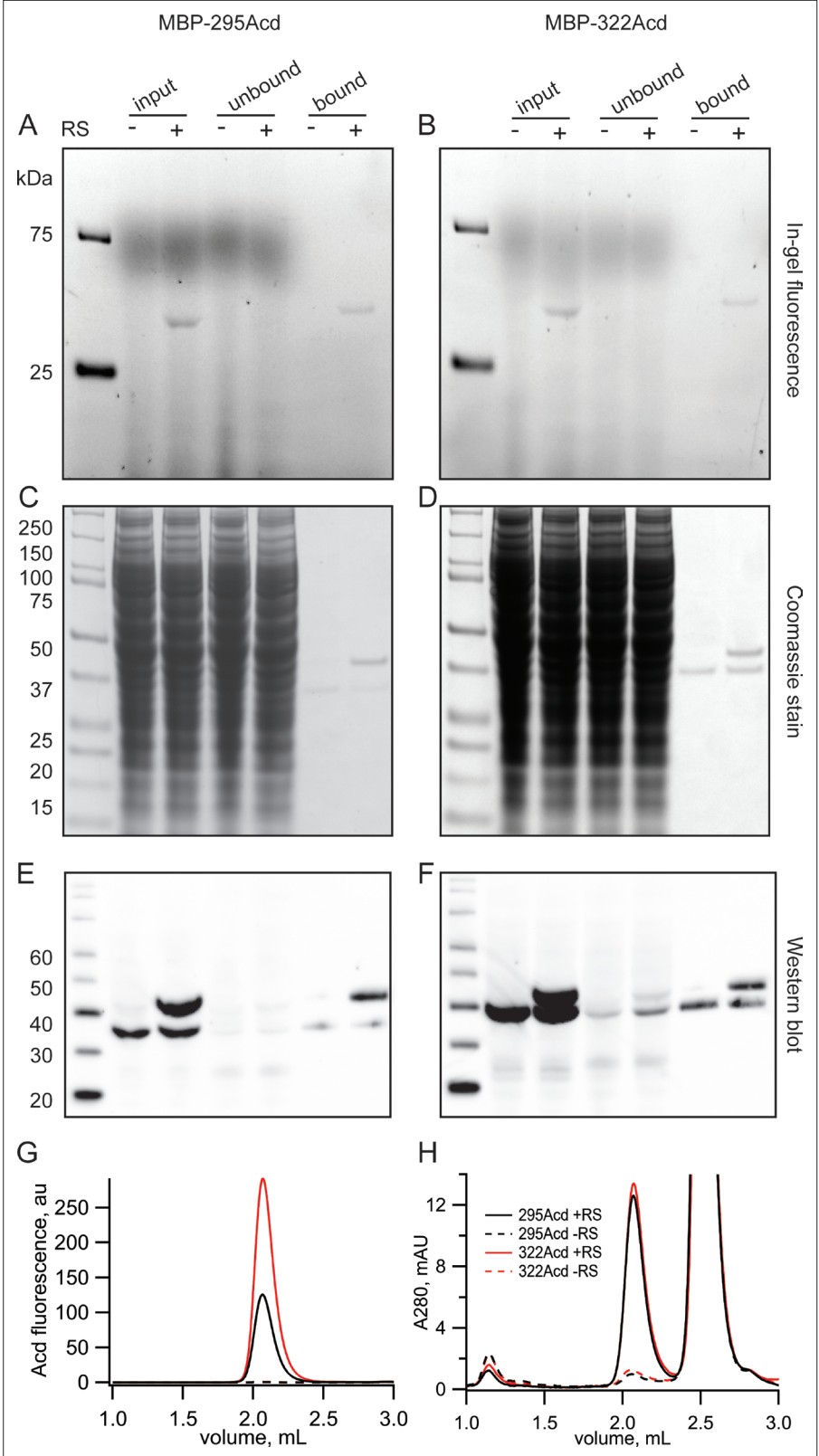

**Figure 2.** Incorporation of Acd into MBP. (**A, B**) In-gel fluorescence for samples of (**A**) MBP-295Acd and (**B**) MBP-322Acd at various points during the purification process. The first lane sample is Precision Plus Dual Stain molecular weight standard. Lanes marked input represent cleared cell lysate. Lanes marked unbound represent the solution removed from the anti-FLAG affinity beads after the binding step. Lanes marked bound represent

*Figure 2 continued on next page*

*Figure 2 continued*

the sample eluted by FLAG peptide. The exclusion and inclusion of the amino acyl tRNA synthetase in the cell transfection is indicated by – and +, respectively. Excitation by a UV transilluminator at 354 nm was used in conjunction with a fluorescein emission filter. (**C, D**) The same gels as shown in (**A, B**) after staining with Coomassie blue. (**E, F**) Western blots using the same samples as (**A, B**) but run on a different gel. Lanes are the same as (**A, B**) except for the first lane in which the MagicMark XP standard was used. (**G, H**) Fluorescence size-exclusion chromatography of purified protein on a Superdex 200 Increase 5/150 GL column. Acd was excited at 385 nm with fluorescence recorded at 450 nm (**G**) and tryptophan absorption was measured at 280 nm (**H**). MBP eluted at 2.1 mL, and the FLAG peptide eluted at 2.5 mL.

The online version of this article includes the following figure supplement(s) for figure 2:

**Source data 1.** In *Figure 2A*, lane 2 is the same as lane 1 shown in the main text figure.

**Source data 2.** In *Figure 2B*, lane 2 is the same as lane 1 shown in the main text figure.

**Source data 3.** In *Figure 2C*, lane 2 is the same as lane 1 shown in the main text figure.

**Source data 4.** In *Figure 2D*, lane 2 is the same as lane 1 shown in the main text figure.

**Source data 5.** In *Figure 2E*, lane 2 is the same as lane 1 shown in the main text figure.

**Source data 6.** In *Figure 2F*, lane 2 is the same as lane 1 shown in the main text figure.

on the backside of the clamshell with β-carbon distances of 13.4 Å in the apo state and 17.6 Å in the holo state, a distance change of +4.2 Å (lengthening). We have previously used these sites on MBP to establish the accuracy of tmFRET with Anap (*Gordon et al., 2018*).

tmFRET was readily observed in both MBP-Acd-C constructs. We recorded Acd emission spectra of purified MBP-295Acd-C and MBP-322Acd-C before and after application of 10 µM $Cu^{2+}$-TETAC. $Cu^{2+}$-TETAC produced a large decrease in fluorescence intensity for both MBP-295Acd-C (*Figure 3C*) and MBP-322Acd-C (*Figure 3D*). The emission spectra in the presence of $Cu^{2+}$-TETAC were nearly identical in shape to the spectra without $Cu^{2+}$-TETAC (*Figure 4C–F*, dashed traces), indicating that the fluorescence quenching reflected a FRET mechanism as opposed to a change of environment of Acd or an inner filter effect. Little or no decrease in fluorescence was observed in constructs lacking the introduced cysteine (MBP-295Acd and MBP-322Acd), indicating that virtually all the quenching was due to energy transfer with $Cu^{2+}$-TETAC conjugated to the introduced cysteine (*Figure 3—figure supplement 1*). Labeling with $Cu^{2+}$-TETAC was nearly complete as shown by the time courses shown in *Figure 4* and the nearly complete quenching observed with a cysteine-reactive quencher with a long $R_0$ value (*Figure 3—figure supplement 2*). Therefore, the fractional decrease in the donor's fluorescence in the presence of acceptor is a measure of the FRET efficiency between the donor and acceptor sites.

The FRET efficiencies for our MBP-Acd constructs were appreciably different in the absence and presence of maltose. For MBP-295Acd-C, the quenching was greater in the presence of maltose, indicating an increase in FRET efficiency and shorter distance (*Figure 3E*). For MBP-322Acd-C, the quenching was smaller in the presence of maltose, indicating a decrease in FRET efficiency and longer distance (*Figure 3F*). These results demonstrate that tmFRET could be used to visualize the conformational change in MBP resulting from the binding of maltose – reporting a decrease in distance for the MBP-295Acd-C FRET pair at the outer lip of the clamshell and an increase in distance for the MBP-322Acd-C FRET pair on the back side of the clamshell (*Table 2*).

To quantitatively compare the FRET efficiencies with Acd and Anap, we measured the time course of the fluorescence at a given wavelength before and after addition of $Cu^{2+}$-TETAC and the reducing agent TCEP. Upon application of 10 µM $Cu^{2+}$-TETAC, the fluorescence rapidly (<10 s) decreased to a new steady-state level for all cysteine-containing MBP-Acd-C and MBP-Anap-C constructs in the absence and presence of maltose (*Figure 4*). This decrease in fluorescence was fully reversed by the reducing agent TCEP (2.5 mM), which breaks the $Cu^{2+}$-TETAC disulfide bond and releases the acceptor, confirming that it was due to modification of the cysteine residue with $Cu^{2+}$-TETAC. A very small, nonreversible, decrease was observed for constructs without cysteine (*Figure 4E and F*). The quenching was corrected for this nonspecific quenching as previously described (*Gordon et al., 2018*) (see Materials and methods) and plotted as $F_{Cys}/F_{no\ Cys}$ (*Figure 4A–D*). For both MBP-295-C and MBP-322-C, the quenching in the Acd constructs was smaller than the quenching in the Anap constructs. This finding is generally consistent with the lower $R_0$ value calculated for the Acd/$Cu^{2+}$-TETAC FRET

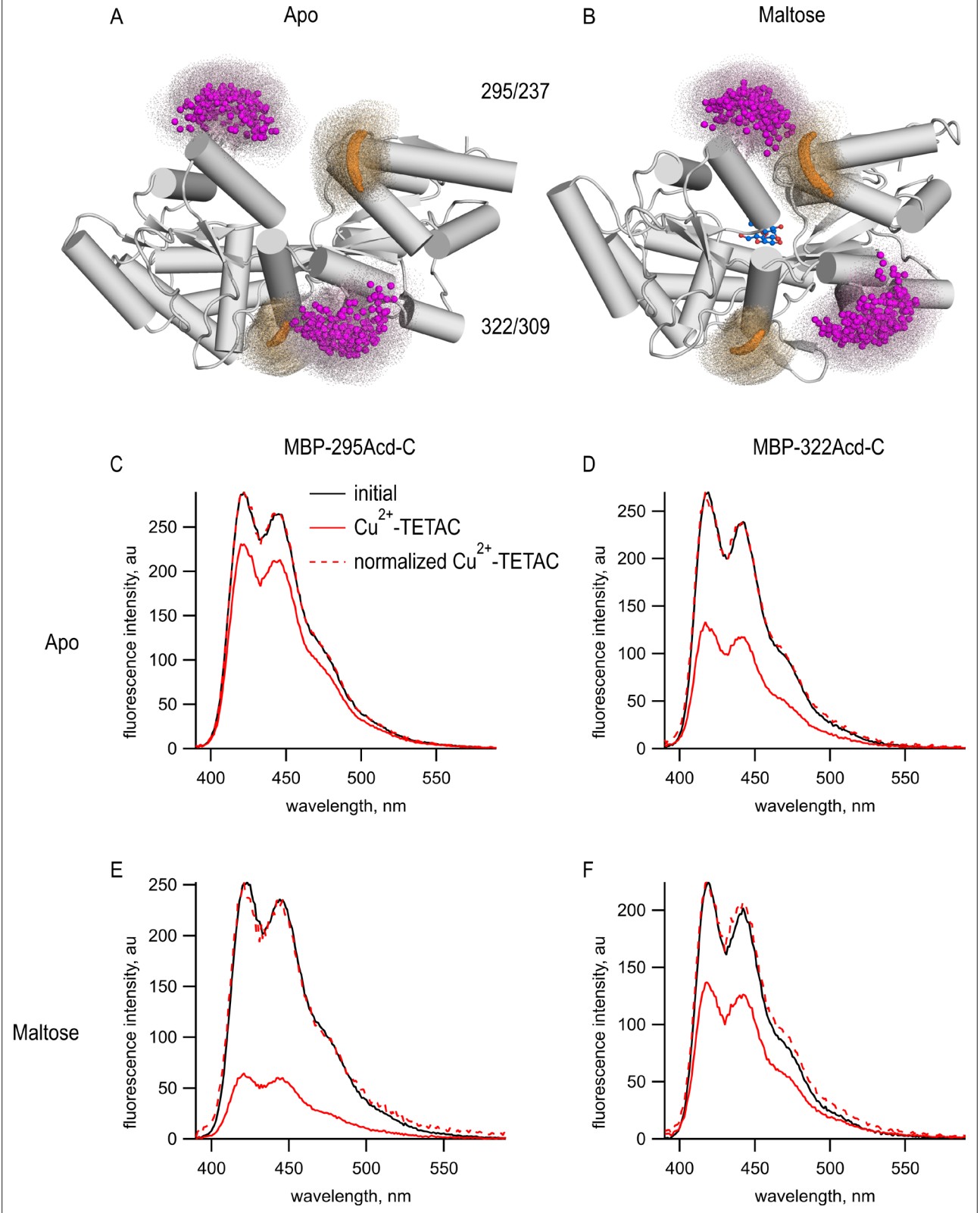

**Figure 3.** tmFRET between Acd incorporated into MBP and Cu²⁺-TETAC. (**A, B**) Cartoon representations of MBP in the (**A**) apo (PDB 1OMP) and (**B**) holo (PDB 1ANF) conformations. The clouds represent possible conformers of Acd or Cu²⁺-TETAC with the carbonyl carbon of the acridone ring of Acd shown in orange and the Cu²⁺ ion of Cu²⁺-TETAC shown in magenta. Predicted donor-acceptor distances are listed in Table 2. (**C–F**) Emission spectra in response to 375 nm excitation, with 5 nm slits on both excitation and emission for (**C, E**) MBP-295Acd-C and (**D, F**) MBP-322Acd-C before (black)

*Figure 3 continued on next page*

*Figure 3 continued*

and after (red) application of Cu²⁺-TETAC. The dashed lines represent the Cu²⁺-TETAC data normalized to the peak initial data. (**C, D**) In the absence of maltose. (**E, F**) In the presence of 10 mM maltose.

The online version of this article includes the following figure supplement(s) for figure 3:

**Figure supplement 1.** Cu²⁺-TETAC did not quench Acd in MBP constructs lacking the cysteine metal-binding site.

**Figure supplement 2.** Quenching of MBP-322Acd by Tide Quencher 1 shows near-complete labeling.

pair (14.9 Å; *Table 2*) than for the Anap/Cu²⁺-TETAC FRET pair (17.2 Å). For MBP-295, the maltose dependence of the FRET was similar for Acd and Anap. For MBP-322, the maltose dependence was lower for Acd than Anap. This probably reflects some small state dependence of the rotameric distribution for Acd or Anap. Overall, however, the tmFRET measurements with Acd and Anap were comparable and indicate that Acd can also be used as a donor for tmFRET.

From the FRET efficiencies, we calculated the distances between the donors and acceptors using the Förster equation $R = R_0(1/E - 1)^{1/6}$. We calculated $R_0$ using the emission spectrum and quantum yield of Acd (0.8) and the absorption spectrum of Cu²⁺ bound to cyclen (*Table 2*). We assumed random orientations of the donor and acceptor ($\kappa^2 = 2/3$), a reasonable assumption when one member of the FRET pair is a metal ion (*Haas et al., 1978*; see Materials and methods). *Figure 4—figure supplement 1* compares the tmFRET distance measurements for MBP-295Acd-C and MBP-322Acd-C with Cu²⁺-TETAC in the absence (open circles) or presence (closed circles) of maltose. Also shown are the β-carbon distances predicted from the X-ray crystal structures of MBP in the absence and presence of ligand. As we have previously demonstrated with MBP-Anap, the experimentally determined distances are generally smaller than the predictions above $R_0$ and larger than the predictions below $R_0$ (*Gordon et al., 2018*). This results in an underestimate of the change in distance due to maltose. This effect can be partially mitigated by using a Förster equation convolved with a Gaussian function (Förster convolved Gaussian [FCG]) for the distance dependence of the FRET efficiency (*Gordon et al., 2018*).

## Lifetime measurements of MBP-Acd

Perhaps the most significant advantage to Acd is that it exhibits long, single-exponential fluorescence lifetimes (*Figure 1E*). The long duration ($\tau_{D_1}$ = 16 ns) provides a large dynamic range for changes in lifetime resulting from FRET, and the single-exponential distribution greatly simplifies the analysis of the time-resolved FRET. Therefore, we measured the fluorescence lifetimes of MBP-295Acd and MBP-322Acd using a frequency-domain lifetime instrument. The lifetime of MBP-322Acd was well fit with a single exponential with an average of 15.1 ns, similar to free Acd (*Table 1*). However, the lifetimes of MBP-295Acd could not be fit by a single exponential, requiring at least two exponentials (*Figure 5A*), with time constants $\tau_{D_1}$ = 14 ns and $\tau_{D_2}$ = 2.0 ns and a relative amplitude $\alpha_1$ = 0.7 (*Figure 5D and E* and *Table 1*).

The shorter, nonsingle-exponential lifetimes observed for MBP-295Acd suggest that something in the environment of Acd is quenching its fluorescence in this construct. This is also consistent with the decrease in specific fluorescence we observed for MBP-295Acd relative to MBP-322Acd (*Figure 2G,H*). Previously, it was shown that Acd can be quenched by nearby tyrosine and tryptophan residues via photoinduced electron transfer (*Speight et al., 2013*). Close examination of the structure of MBP revealed a likely culprit for the quencher in MBP-295Acd. Tyrosine at position 307 is predicted to be in close proximity (<5 Å) to Acd at position 295 (*Figure 5B*). Indeed, the mutation Y307F increased the average lifetime to 15.5 ns and completely eliminated the double exponential character of MBP-295Acd (*Figure 5C,D* and *Table 1*). The MBP-295Acd-Y307F-C construct maintained the large maltose dependence of tmFRET, although the steady-state FRET efficiency was somewhat lower than MBP-295Acd-C, perhaps because Y307F altered the rotameric distribution of Acd (*Figure 5—figure supplement 1*). For the remainder of this study, every construct with Acd incorporated at position 295 also included the Y307F mutation.

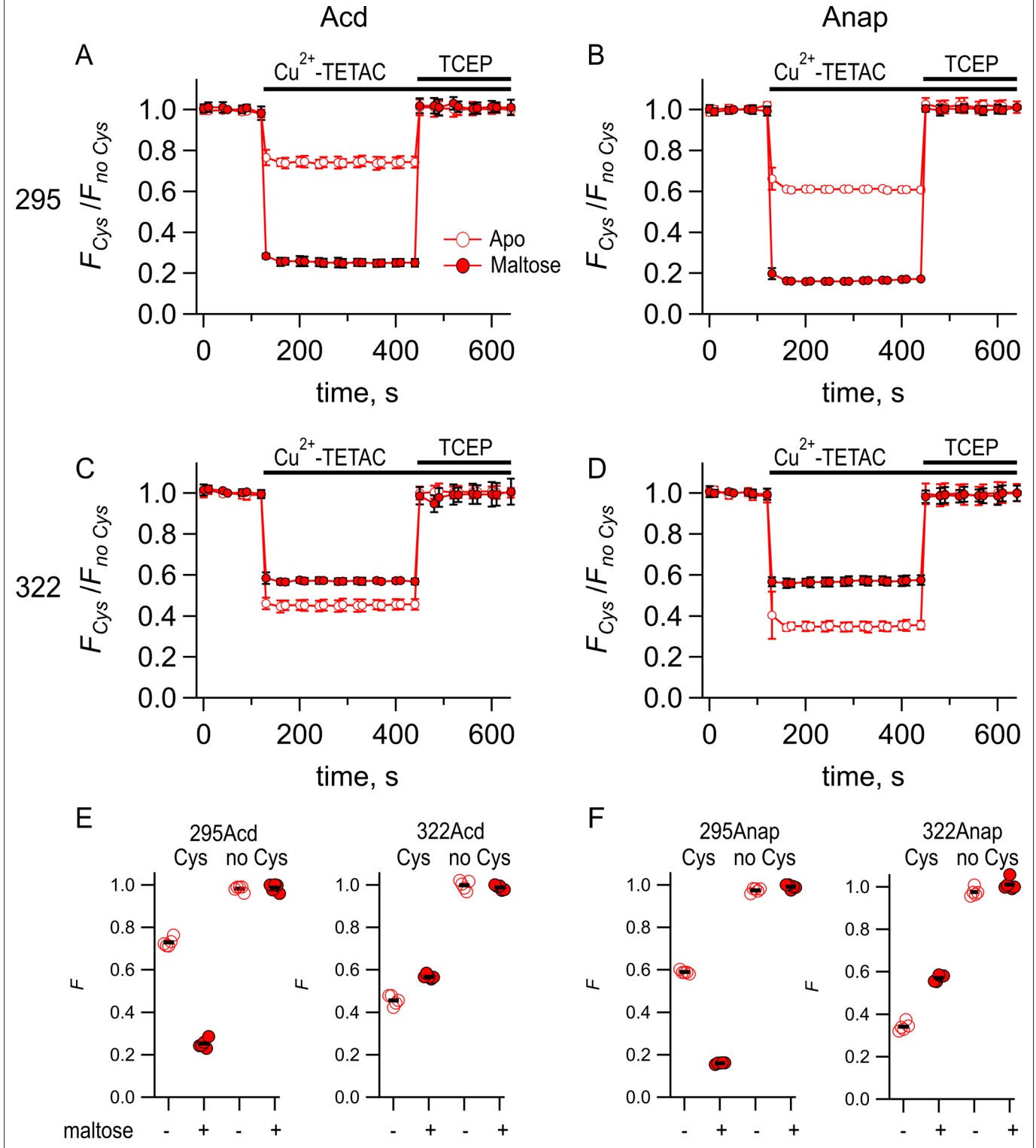

**Figure 4.** Comparing Acd and Anap as donors for transition metal ion fluorescence resonance energy transfer (tmFRET). Quenching by $Cu^{2+}$-TETAC and recovery with TCEP for (**A**) MBP-295Acd-C, (**B**) MBP-295Anap-C, (**C**) MBP-322Acd-C, and (**D**) MBP-322Anap-C, with $F_{Cys}$ and $F_{no\ Cys}$ as defined in **Equation 2**. Data are shown as mean ± SEM for n = 5. 10 µM $Cu^{2+}$-TETAC and 2.5 mM TCEP were added at the times indicated by the bars. Open symbols represent data collected in the absence of maltose, and filled symbols represent data collected in the presence of 10 mM maltose. (**E–F**) Collected data showing each independent sample (circles) and the mean (black line) for $F_{Cys}$ and $F_{no\ Cys}$ for (**E**) MBP-295Acd-C (labeled 295Acd Cys) and MBP-295Acd (labeled 295Acd no Cys), MBP-322Acd-C (labeled 322Acd Cys) and MBP-322Acd (labeled 322Acd no Cys) and for (**F**) MBP-295Anap-C (labeled 295Anap

*Figure 4 continued on next page*

*Figure 4 continued*

Cys) and MBP-295Anap (labeled 295Anap no Cys), MBP-322Anap-C (labeled 322Anap Cys) and MBP-322Anap (labeled 322Anap no Cys). The absence and presence of 10 mM maltose are as indicated by the – and + shown beneath each data set.

The online version of this article includes the following figure supplement(s) for figure 4:

**Figure supplement 1.** Comparing distances determined from experimental data to predicted distances from X-ray structures.

## Lifetime measurements of tmFRET reveal heterogeneity of donor-acceptor distances

Previously, we determined the apparent efficiency of tmFRET with Anap using only steady-state measurements (*Gordon et al., 2018*), as also described above for Acd. The FRET efficiencies did not conform to the sixth-power distance dependence predicted by the Förster equation, but, instead, exhibited a shallower distance dependence. This shallower distance dependence was well fit with a Förster equation convolved with a Gaussian function (FCG). The FCG equation is the prediction of the apparent FRET efficiency if the distance between the donor and acceptor was heterogeneous instead of fixed. This heterogeneity in each conformational state was described by a Gaussian distribution, with a mean distance, $\bar{r}$, and standard deviation, $\sigma$, where the distances did not change appreciably on the time scale of the fluorescence lifetime (*Wozniak et al., 2008*). This model also assumed that $\sigma$ was the same for all the states and constructs that produced different mean distances, a simplifying assumption required when using steady-state FRET but which is not physically justified.

The long, single-exponential lifetime of Acd allowed us to measure the heterogeneity of donor-acceptor distances more directly than was possible with the FCG approach. FRET causes a decrease in the fluorescence lifetime of the donor by introducing a nonradiative pathway for the donor fluorophore to relax from its excited state, which can then be used to calculate the FRET efficiency. A continuous distribution of distances would produce a continuous distribution of lifetimes that can be fit to the lifetime data in either the time domain or the frequency domain. Like FCG, the approach assumes a particular form for the distance distribution, such as a Gaussian (*Lakowicz et al., 1994a*) or sum of Gaussians (*Kulinski et al., 1997*), and that the distances do not change appreciably on the time scale of the fluorescence lifetime (*Wozniak et al., 2008*). A major advance in using lifetimes instead of steady-state approaches, however, is that $\sigma$ need not be the same for different constructs/conditions. Because the distributions are parameterized, the fits generally have fewer free parameters than fitting with a sum of exponentials. In addition, the use of structure-based parameters (e.g., $\bar{r}$ and $\sigma$) allows the distances to be more easily interpreted in terms of the molecular structures. This approach was pioneered in the 1970s, mostly by Steinberg and coworkers (*Grinvald et al., 1972*; *Haas et al., 1975*).

We measured tmFRET with fluorescence lifetimes to determine the heterogeneity of the donor-acceptor distances in our MBP-Acd constructs. For each protein sample, we measured the fluorescence lifetime in the donor-only condition, after the addition of 10 μM Cu$^{2+}$-TETAC and after the addition of 18 mM TCEP. For each condition, phase delay, $\varphi_\omega$, and modulation ratio, $m_\omega$, at each modulation frequency, $\omega$, along with the steady-state quenching data, $E$, were simultaneously fit with a model for the fluorescence lifetimes assuming a Gaussian distribution of distances between the donor

**Table 2.** Predicted donor-acceptor distances from MBP X-ray crystal structures.

| Donor | Acceptor | $R_0$ (Å) | Apo vs. Holo | $r_{predicted,}$ β-carbon (Å) | $r_{predicted,}$ centroid (Å) |
|---|---|---|---|---|---|
| *MBP-295Acd-Y307F-C* | Cu$^{2+}$-TETAC | 14.9 | Apo | 21.3 | 21.7 |
| *MBP-295Acd-Y307F-C* | Cu$^{2+}$-TETAC | 14.9 | Holo | 12.8 | 12.1 |
| *MBP-295Acd-Y307F-HH* | Cu$^{2+}$ | 12.2 | Apo | 18.7 | 17.1 |
| *MBP-295Acd-Y307F-HH* | Cu$^{2+}$ | 12.2 | Holo | 10.9 | 7.8 |
| *MBP-322Acd-C* | Cu$^{2+}$-TETAC | 14.9 | Apo | 13.4 | 11.2 |
| *MBP-322Acd-C* | Cu$^{2+}$-TETAC | 14.9 | Holo | 17.7 | 18.8 |

$r_{predicted,}$ β-carbon, and $r_{predicted}$ centroid were determined from PDB structures 1OMP and 1ANF. The calculated $R_0$ values and predicted distances for MBP-295Acd constructs are the same as for MBP-295Acd-Y307F constructs.

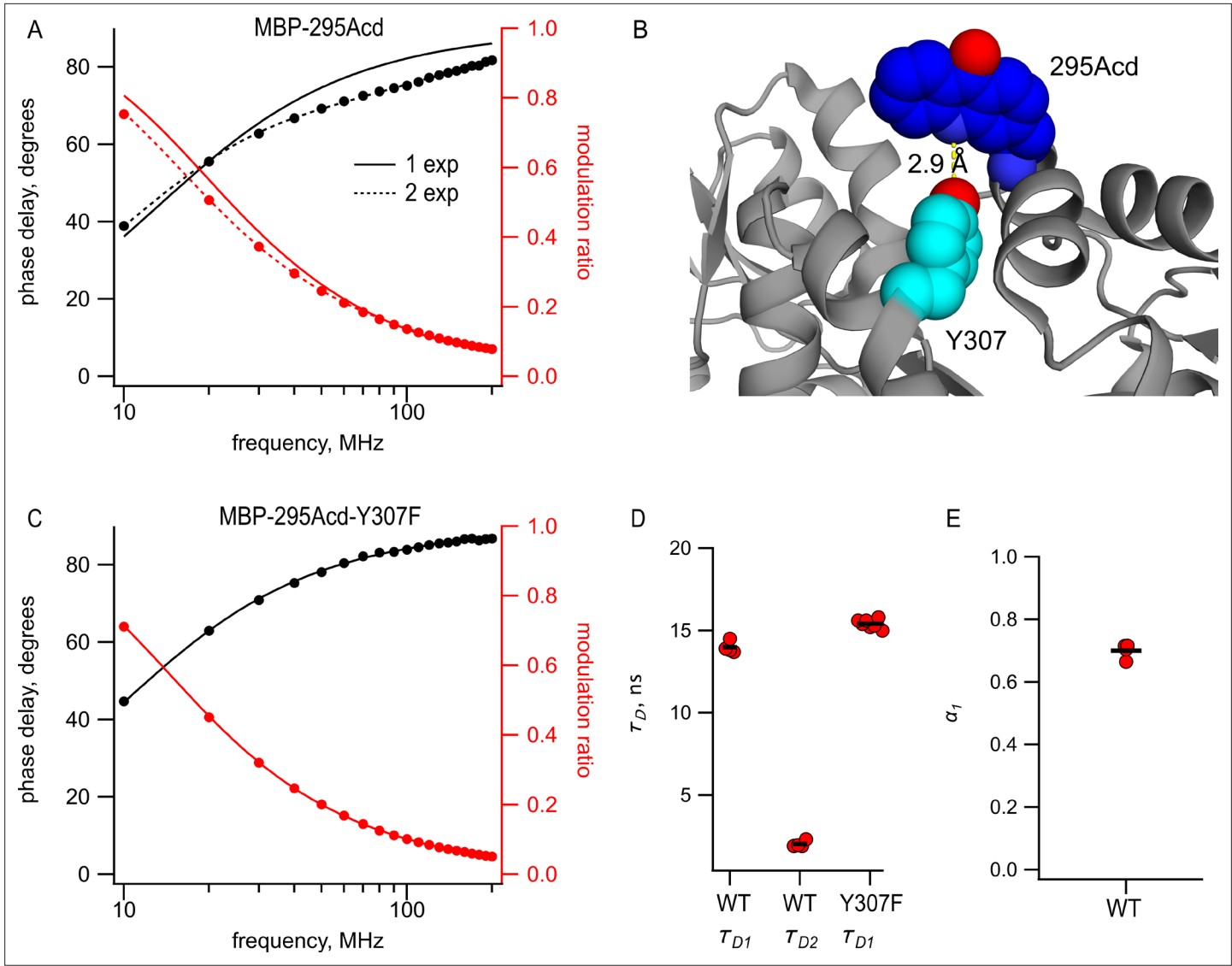

**Figure 5.** Acd in MBP-295Acd is nonexponential and introduction of the Y307F mutation increases its lifetime and makes it single exponential. (**A, C**) Frequency-domain measurements of fluorescence lifetime of (**A**) MBP-295Acd and (**C**) MBP-295Acd-Y307F. Black, phase delay in degrees; red, modulation ratio. Curves are fits with *Equations 5* and *6* corrected by *Equations 13* and *14*. Fit parameters for (**A**) are $\tau_{D_1}$ = 11.6 ns for a single exponential (solid curve) and $\tau_{D_1}$ = 13.9 ns, $\tau_{D_2}$ = 1.9 ns, and $\alpha_1$ = 0.70 Å (dashed curve) and for (**C**) are $\tau_{D_1}$ = 15.6 ns. (**B**) Cartoon representation of the proximity between Acd at position 295 and tyrosine at position 307 in the apo state. (**D, E**) Collected data from fits for MBP-295Acd (WT) and MBP-295Acd-Y307F, as indicated. Independent samples are shown as red circles, and the means are shown as black lines.

The online version of this article includes the following figure supplement(s) for figure 5:

**Figure supplement 1.** Steady-state fluorescence to measure transition metal ion fluorescence resonance energy transfer (tmFRET) between MBP-295Acd-Y307F-C and $Cu^{2+}$-TETAC.

and acceptor (see Materials and methods). The values of 3–4 free parameters for each condition were determined using $\chi^2$ minimization.

The model with Gaussian-distributed distances provided an excellent fit to the lifetime data for MBP-295Acd-Y307F-C and MBP-322Acd-C in the absence and presence of maltose. The data for the donor-only condition for MBP-295Acd-Y307F-C and MBP-322Acd-C were well fit by single-exponential lifetimes (*Figure 6A–D*, filled circles, and *Table 1*). The addition of $Cu^{2+}$-TETAC caused a shift in the phase delay and modulation ratio data to higher frequencies, indicative of a shortening of the fluorescence lifetime of the donor (*Figure 6A–D*, bowties, and *Table 1*). For MBP-295Acd-Y307F-C, the shift was larger in the presence of maltose than in the absence of maltose (*Figure 6A,C*, filled vs. open

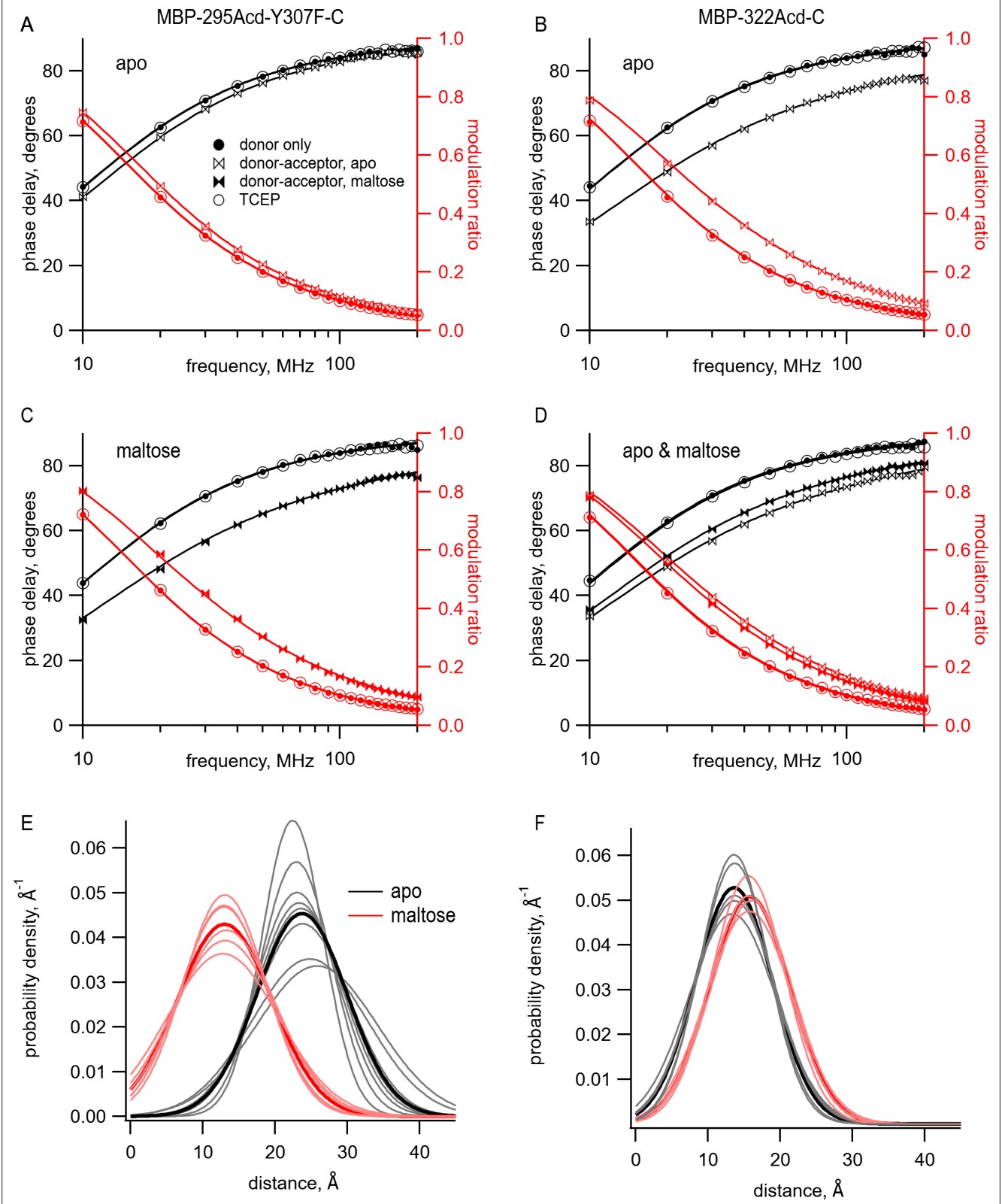

**Figure 6.** Using time-resolved transition metal ion fluorescence resonance energy transfer (tmFRET) to measure distance and distance distributions between Acd and Cu²⁺ bound to di-histidine in MBP. (**A–D**) Frequency-domain measurements of fluorescence lifetime of (**A, C**) MBP-295Acd-Y307F-C and (**B, D**) MBP-322Acd-C. Black, phase delay in degrees; red, modulation ratio. Filled circles represent the donor-only condition at the beginning of the experiment. For (**A, B, D**), this was in the absence of maltose and for (**C**) this was in the presence of 10 mM maltose. Open bowties represent

*Figure 6 continued on next page*

*Figure 6 continued*

the addition of 10 µM Cu$^{2+}$-TETAC in the absence of maltose. Filled bowties represent the addition of 10 µM Cu$^{2+}$-TETAC in the presence of 10 mM maltose. Open circles represent data collected after sequential addition of TCEP. Solid curves represent fits to the data with *Equations 5* and *6* and then corrected using *Equations 13* and *14*. The parameters for the fits shown in the figure are as follows: (**A**) $\tau_{D_1} = 15.5$ ns, $\bar{r}_1 = 22.9\,\mathring{A}$, $\sigma_1 = 5.0\,\mathring{A}$; (**B**) $\tau_{D_1} = 15.2$ ns, $\bar{r}_1 = 13.7\,\mathring{A}$, $\sigma_1 = 5.7\,\mathring{A}$; (**C**) $\tau_{D_1} = 15.1$ ns, $\bar{r}_2 = 12.8\,\mathring{A}$, $\sigma_2 = 7.8\,\mathring{A}$; and (**D**) $\tau_{D_1} = 15.4$ ns, $\bar{r}_1 = 13.7\,\mathring{A}$, $\sigma_1 = 5.0\,\mathring{A}$, $\bar{r}_2 = 15.7\,\mathring{A}$, $\sigma_2 = 5.9\,\mathring{A}$. (**E, F**) Spaghetti plots representing the center and width of the Gaussians used to fit the distance distributions for each experiment (thin curves) and the average of all fits (thick curves) for (**E**) MBP-295Acd-Y307F-C and (**F**) MBP-322Acd-C in either the absence (black) or presence (red) of 10 mM maltose.

The online version of this article includes the following figure supplement(s) for figure 6:

**Figure supplement 1.** Cu$^{2+}$-TETAC did not alter the lifetimes of constructs that did not contain the metal-binding site.

**Figure supplement 2.** Predictions of distance distributions from mtsslWizard.

**Figure supplement 3.** Illustrations of distance distributions from mtsslWizard.

bowties). This is consistent with a maltose-dependent decrease in distance, which would be expected to decrease the lifetimes for the MBP-295Acd-Y307F-C construct. In contrast, for MBP-322Acd-C, the shift was smaller in the presence of maltose than in the absence of maltose (*Figure 6D*, filled vs. open bowties, and *Table 1*), consistent with the maltose-dependent increase in distance expected for the MBP-322Acd-C construct. In all cases, the data in the presence of acceptor were not well fit by a single-exponential lifetime distribution, as expected for a single FRET distance, but were well fit by the model with Gaussian-distributed distances. And, in all cases, the reductions in lifetime reversed nearly completely after application of TCEP (*Figure 6A–D*, open circles, and *Table 1*). No detectable changes in lifetime occurred for similar experiments using MBP-295Acd-Y307F or MBP-322Acd lacking an acceptor-site cysteine (*Figure 6—figure supplement 1*).

The distribution of distances predicted from the fits to the data in the presence of acceptor is shown for multiple samples in *Figure 6E,F* with parameter values given in *Table 1*. Both the mean distances, $\bar{r}$, and the standard deviations, $\sigma$, of the distributions were fairly consistent across samples. The largest $\sigma$ occurred with MBP-295Acd-Y307F-C in the absence of maltose (*Figure 6E*, black traces), the condition with the longest predicted distance and least amount of FRET. For MBP-295Acd-Y307F-C, the $\bar{r}$ decreased from 23.7 Å to 13.4 Å in the presence of maltose, more consistent with the decrease in distance observed in the X-ray crystal structures than measured using steady-state tmFRET (*Figure 4—figure supplement 1*). For MBP-322-C, $\bar{r}$ increased from 13.6 Å to 15.8 Å in the presence of maltose, somewhat smaller than the increase in distance in the X-ray crystal structures (*Figure 4—figure supplement 1*). Interestingly, the estimated $\sigma$ with Cu$^{2+}$-TETAC for all conformational states were quite large. In the absence of maltose, $\sigma$ values were 6.2 Å for MBP-295Acd-Y307F-C and 5.3 Å for MBP-322-C (corresponding to a FWHM of 14.6 Å and 12.5 Å, respectively) and in the presence of 10 mM maltose, $\sigma$ values were 6.5 Å for MBP-295Acd-Y307F-C and 5.5 Å for MBP-322-C (corresponding to a FWHM of 15.3 Å and 13.0 Å). The widths of these distributions are generally consistent with the predicted donor-acceptor distance distributions considering all possible rotamers of Acd and Cu$^{2+}$-TETAC attached to a cysteine residue (*Figure 6—figure supplement 2A,B*). The heterogeneity in the distances likely reflects the relatively long linker associated with the acceptor Cu$^{2+}$-TETAC attached to a cysteine residue.

## Lifetime measurements with tmFRET reveal distributions among conformational states

The model for fluorescence lifetimes with Gaussian-distributed distances can reveal not only the heterogeneity of the distances within any given conformational state, $\sigma$, but also the distribution among conformational states (parameter $A_2$ in Figure 8C ). Knowing the distribution among states at equilibrium allows us to calculate an equilibrium constant, and therefore a free energy difference, between states. Thus, time-resolved FRET experiments have the capability to reveal sparse structural information on intramolecular distances and heterogeneity and to quantify the energetics of conformational rearrangements within proteins or protein domains.

To determine whether tmFRET with Acd has the capability to measure the distribution between conformational states of MBP, we measured tmFRET in MBP-295Acd-Y307F-HH at subsaturating concentrations of maltose. For these experiments, we used Cu$^{2+}$ bound to a di-histidine motif as an acceptor instead of Cu$^{2+}$-TETAC. This change was motivated by the relatively large rotameric

cloud expected for $Cu^{2+}$-TETAC, illustrated in **Figure 3A,B**, to which we attribute the large $\sigma$ values described above (**Figure 6E,F**). For the MBP-295Acd-Y307F-HH construct, we introduced the di-histidine motif near the acceptor site in MBP-295Acd-C, with the histidines at positions 233 and 237 separated by one turn on an α-helix. We have previously shown, using steady-state tmFRET in MBP-295Anap-HH, that this di-histidine motif creates minimal metal-binding sites for $Cu^{2+}$ and other transition metals, with an apparent affinity for $Cu^{2+}$ of 3 µM (**Gordon et al., 2018**). The fluorescence of MBP-295Acd-Y307F-HH was quenched upon application of a saturating concentration (100 µM) of $Cu^{2+}$ and recovered upon addition of ethylenediaminetetraacetic acid (EDTA; 10 mM), indicative of tmFRET between the Acd and $Cu^{2+}$ bound to the di-histidine (**Figure 7—figure supplement 1**). Furthermore, as expected, MBP-295Acd-Y307F-HH exhibited a large increase in FRET efficiency (decrease in distance) with the maltose-dependent closure of the MBP clamshell. Converting the FRET efficiencies to distances revealed some underprediction of the maltose-dependent distance change, as seen for $Cu^{2+}$-TETAC (**Figure 4—figure supplement 1**).

To determine the distance distributions in MBP-295Acd-Y307F-HH at different concentrations of maltose, we measured tmFRET using fluorescence lifetimes. For each protein sample, we first measured the fluorescence lifetime of the donor-only in the absence of acceptor (**Figure 7A**, gray). The data were well fit with a single-exponential lifetime (**Table 1**) of 15.6 ns, which was not significantly different from what we observed for MBP-295Acd-Y307F-C (Student's t-test, two-tailed, p=0.35). We then added 100 µM $Cu^{2+}$ and measured tmFRET in five different conditions: (1) no maltose (red), (2) 200 µM maltose (green), (3) 370 µM maltose (blue), (4) 10 mM maltose (saturating maltose; magenta), and (5) after the addition of 17 mM EDTA to remove the $Cu^{2+}$ from the di-histidine-binding site. As shown in **Figure 7A**, addition of 100 µM $Cu^2$ caused a shift of the phase delay and modulation ratio to higher frequencies, indicative of a decrease in lifetime and FRET between Acd and $Cu^{2+}$ bound to the di-histidine motif of MBP-295Acd-Y307F-HH. Addition of increasing concentrations of maltose produced further shifts in the phase delay and modulation ratio. This effect was almost completely reversed by the application of EDTA (**Table 1**). These results indicate that MBP-295Acd-Y307F-HH undergoes a maltose-dependent increase in average FRET efficiency, and subsaturating concentrations of maltose produced intermediate apparent FRET efficiencies.

We first fit the data in the absence of maltose and in the presence of 10 mM maltose, as described above. These data could be well fit by our FRET model with a single Gaussian distance distribution (**Figure 7A** and **Table 1**). The mean distance $\bar{r}$ decreased from $\bar{r}_1$ = 18.3 Å to $\bar{r}_2$ = 12.7 Å, more consistent with the maltose-dependent change in distance from the X-ray crystal structures than the distance change determined with steady-state FRET (**Figure 7C** and **Figure 4—figure supplement 1**). Interestingly, the $\sigma$ values ($\sigma_1$ = 2.5 Å and $\sigma_2$ = 1.3 Å in the absence and presence of 10 mM maltose, respectively) were substantially smaller than the $\sigma$ values when $Cu^{2+}$-TETAC was used as the acceptor ($\sigma_1$ = 6.2 Å and $\sigma_2$ = 6.5 Å). This finding likely reflects that $Cu^{2+}$ bound to a di-histidine motif is more rigid and closely associated with the protein backbone than $Cu^{2+}$-TETAC attached to a cysteine residue, as seen in the rotameric clouds of the $Cu^{2+}$-di-histidine (**Figure 6—figure supplement 3**) and predicted distance distributions of all possible rotamers (**Figure 6—figure supplement 2C**). Moreover, the value of σ in the absence of maltose was larger than in the presence of 10 mM maltose, consistent with previous findings from NMR, small-angle X-ray scattering (SAXS), and EPR spectroscopy (**Tang et al., 2007**; **Selmke et al., 2018**). The greater heterogeneity in distances in the absence vs. presence of maltose likely reflects increased heterogeneity in the degree to which the MBP clamshell is open in the apo state (**Tang et al., 2007**). These results highlight the power of using time-resolved FRET to measure distance distributions.

To determine whether time-resolved FRET with Acd could be used to quantify the distribution among multiple conformational states, we analyzed the fluorescence lifetime data at 200 µM and 370 µM maltose. For these fits, we assumed that the distance distributions were described by the sum of two Gaussians, one with $\bar{r}_1$ and $\sigma_1$ of the open clamshell state and one with $\bar{r}_2$ and $\sigma_2$ of the closed clamshell state (see **Figure 8C**). The only parameter in the distance distribution that was allowed to vary was the proportion of the open vs. closed clamshell states ($A_2$). **Figure 7A** shows that these distributions provided excellent fits to the fluorescence lifetime data at subsaturating maltose concentrations. Based on the relative area under each Gaussian for the clamshell closed vs. open populations, the proportion of the maltose-bound (clamshell closed) conformation was 0.38 in 200 µM maltose and 0.60 in 370 µM maltose (**Table 1**). These proportions closely matched the predictions based on

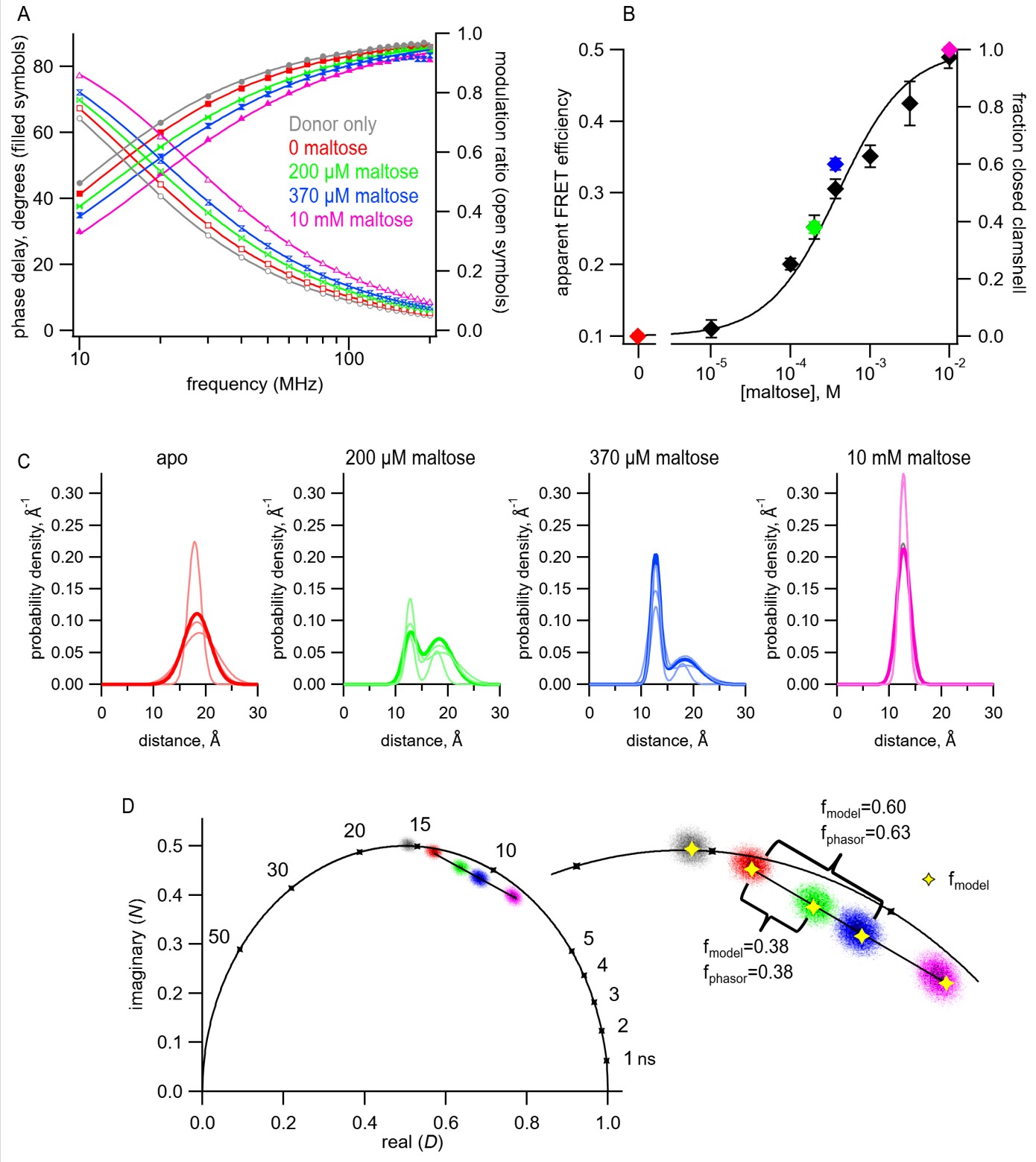

**Figure 7.** Transition metal ion fluorescence resonance energy transfer (tmFRET) between Acd and Cu²⁺ in MBP can be used to measure distributions among conformations at subsaturating maltose concentrations. (**A**) Frequency-domain measurements of fluorescence lifetimes of MBP-295Acd-HH measured sequentially in the absence of acceptor (donor only – gray), 100 μM Cu²⁺ (red), 100 μM Cu²⁺ and 200 μM maltose (green), 100 μM Cu²⁺ and 370 μM maltose (blue), and 100 μM Cu²⁺ and 10 mM maltose (magenta). Filled symbols represent phase delay, and open symbols represent modulation ratio. Solid curves are fits with *Equations 5* and *6* corrected with *Equations 13* and *14* with the following fit parameters:

*Figure 7 continued on next page*

*Figure 7 continued*

$\tau_{D_1} = 15.7$ ns, $\bar{r}_1 = 18.8$ Å, $\sigma_1 = 3.5$ Å, $\bar{r}_2 = 12.6$ Å and $\sigma_2 = 1.3$ Å. (**B**) The dependence of apparent FRET efficiency measured with steady-state FRET (black) or fraction in the closed clamshell conformation measured with time-resolved FRET (red) on maltose concentration. Points are mean ± SEM for 3–5 independent experiments. The fraction of the population in the closed clamshell conformation at each MBP concentration was taken from model fits to the data, as shown in (**A**). Solid curve is a fit with the Hill equation with the following parameters: $K_{1/2} = 393$ μM and Hill slope = 1. (**C**) Spaghetti plots representing the center and width of the Gaussians used to fit the distance distributions for each experiment (thin curves) and the average of all fits (thick curves) for MBP-295Acd-HH at the indicated maltose concentrations. (**D**) Phasor plot in which each pixel of the fluorescence lifetime imaging microscopy image is plotted as a dot, with the color corresponding to the maltose concentration, as in (**A**) and (**C**). The line connects the centroid of the data in the absence of maltose to that in the presence of 10 mM maltose. (Inset) Magnified section of the phasor plot. The yellow points represent $N$ and $D$ predicted from *Equations 7* and *8* from fits of the model to the frequency-domain data shown in (**A**). The fraction of the population in the closed clamshell conformation predicted by the models is compared to that given by the phasor plots.

The online version of this article includes the following figure supplement(s) for figure 7:

**Figure supplement 1.** Steady-state measurement of transition metal ion fluorescence resonance energy transfer (tmFRET) between MBP-295Acd-Y307F-HH and $Cu^{2+}$.

the apparent FRET efficiency at subsaturating maltose concentrations measured from steady-state quenching experiments (*Figure 7B*). The maltose apparent affinity from both steady-state and lifetime FRET experiments was ~400 μM, similar to the 280 μM maltose affinity determined previously for MBP-295Anap-C with the W340A mutation but without the Y307F mutation (*Gordon et al., 2018*) and to the ~1 mM $K_D$ reported for MBP (*Martineau et al., 1990*).

For fluorescence lifetime imaging microscopy (FLIM) experiments, the fluorescence lifetime data are sometimes displayed in a representation that highlights the different lifetime signatures in a model-independent way. These 'phasor plots' are polar plots of the raw phase delay and modulation ratio data, or correspondingly Cartesian plots of the imaginary ($N$) vs. the real ($D$) components of the fluorescence response, for each pixel at a given modulation frequency (*Figure 7D*; *Digman et al., 2008*). No assumption is made about the number of decay rates present, the specific models for the decay (e.g., exponential, nonexponential), or the shape of the distance distribution. The pixels that have the same lifetime signature present as a cluster of points in the plot. The clusters that fall on the 'universal circle' conform to a single-exponential fluorescence decay, and those inside the circle are multiexponential or nonexponential.

Phasor plots of the fluorescence lifetime data measured at 10 MHz at different concentrations of maltose allowed us to estimate the distribution of MBP in the two different conformational states, clamshell open and closed, in a model-independent way. As expected, the donor-only data (*Figure 7D*, gray points) fall on the universal circle as they are single-exponentially distributed, whereas the data in the presence of acceptor fall inside the circle (*Figure 7D*, red, green, blue, and magenta). Importantly, any conditions that have a combination of two different lifetime signatures will fall on a straight line that connects the two parent populations. As shown in *Figure 7D*, the data with subsaturating maltose concentrations (green and blue) fall on a straight line connecting the 0 maltose (apo) data (red) and the saturating maltose data (magenta). Moreover, the fractional distance along the line corresponds to the proportion of the individual populations. This model-independent estimate of the proportion of the maltose-bound (clamshell closed) conformation was 0.38 in 200 μM maltose and 0.63 in 370 μM maltose, very similar to the 0.38 and 0.60, respectively, estimated from our FRET model with Gaussian-distributed distances. This analysis provides a model-independent way to estimate the proportion in each conformational states and validates our FRET model with Gaussian-distributed distances (*Digman et al., 2008*).

## Discussion

This paper describes a significant expansion of the tmFRET method. Using MBP as a model system, we demonstrated the incorporation of the fluorescent noncanonical amino acid Acd in mammalian cells using a recently engineered RS (*Jones et al., 2021*). We showed that Acd can be used as a tmFRET donor with $Cu^{2+}$-TETAC or $Cu^{2+}$-di-histidine as the acceptor. We demonstrated that the long, single-exponential fluorescence lifetime of Acd is suitable for time-resolved measurements of the tmFRET efficiency. We presented a FRET model with Gaussian-distributed distances for analyzing the fluorescence lifetime data that allowed calculation of the heterogeneity of distances for each conformational

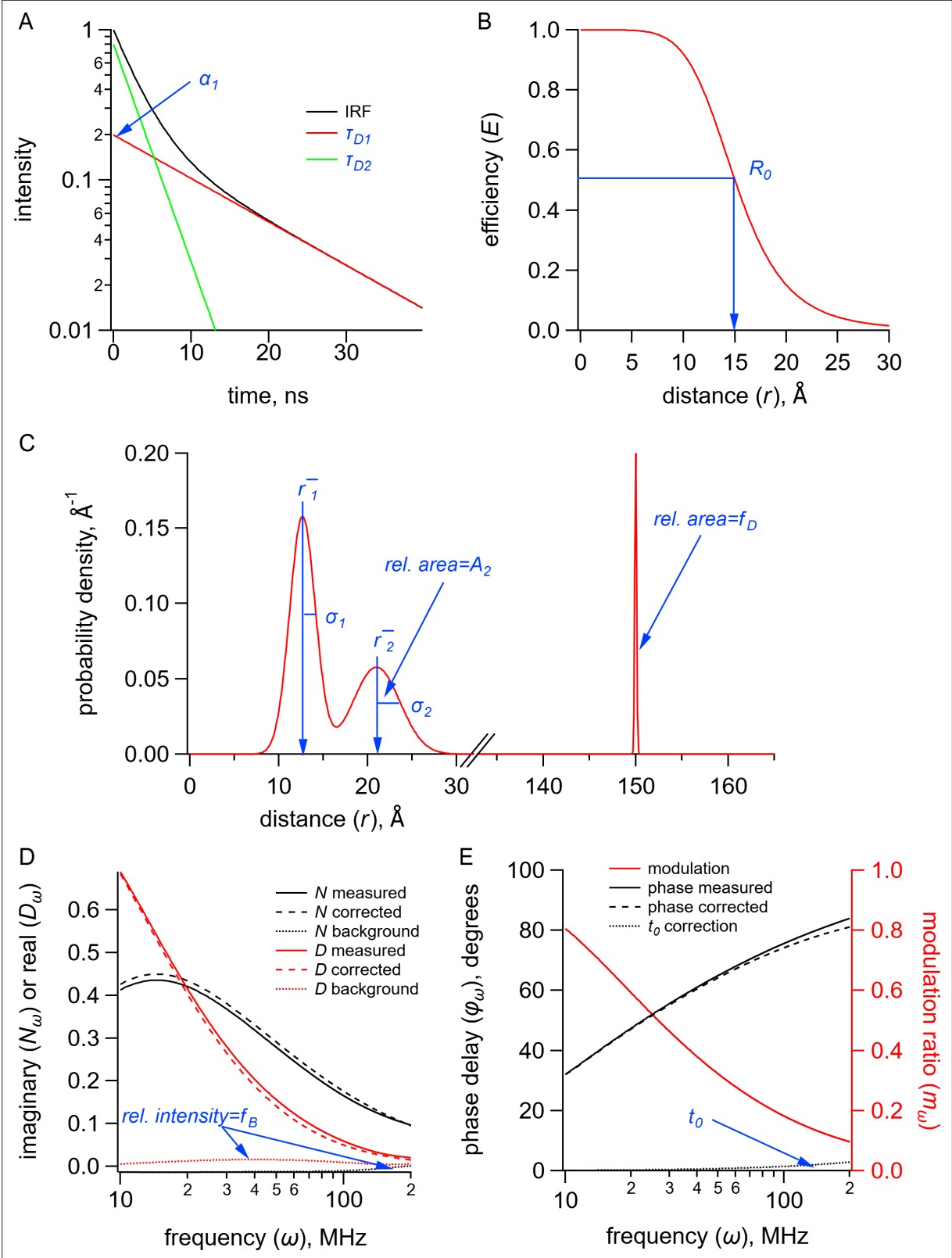

**Figure 8.** Quantities (black) and parameters (blue) in the model for time-resolved transition metal ion fluorescence resonance energy transfer (tmFRET). (**A**) Plot of fluorescence lifetimes in the time domain for a donor fluorophore with two exponential components with time constants ($\tau_{D_1}$ and $\tau_{D_2}$) and relative amplitude ($\sigma_1$). (**B**) Plot of the FRET efficiency (E) as a function of distance ($r$) showing the characteristic distance for the donor-acceptor pair ($R_0$). (**C**) Plot of a distribution of donor-acceptor distances $P(r)$ with two Gaussian components with means ($\bar{r}_1$ and $\bar{r}_2$), standard deviations ($\sigma_1$ and $\sigma_2$),

*Figure 8 continued on next page*

*Figure 8 continued*

and relative amplitude of the second component ($A_2$). The fraction of donor only ($f_D$) was modeled as a narrow Gaussian with a mean distance of 150 Å and a standard deviation of 0.1 Å, too far to exhibit any detectable FRET. (**D**) Plot of the imaginary ($N_\omega$) and real ($D_\omega$) components of the measured, corrected, and background fluorescence response as a function of the modulation frequency ($\omega$) where $f_B$ is the fraction of the fluorescence intensity due to background. (**E**) Plot of the phase delay ($\varphi_\omega$) and modulation ratio ($m_\omega$) of the measured and corrected fluorescence response as a function of the modulation frequency ($\omega$) where $t_0$ is the time shift of the instrument response function.

The online version of this article includes the following figure supplement(s) for figure 8:

**Figure supplement 1.** Spectral overlap between donors and acceptors for Acd experiments.

**Figure supplement 2.** Measuring fluorescence resonance energy transfer (FRET) in the time domain and frequency domain.

state as well as the distribution among different conformational states. This model provides a more direct measure of the heterogeneity than our previous FCG model (*Gordon et al., 2018*) and does not rely on assuming the same heterogeneity in different conditions. Finally, phasor plot analysis produced model-independent measurements of the distribution among conformational states consistent with those of our model, further validating our time-resolved tmFRET approach.

Perhaps the biggest advantage of Acd over Anap is its utility for time-resolved FRET experiments. A standard macroscopic steady-state FRET experiment provides just a single number, the apparent FRET efficiency, from which one can calculate a single weighted-average distance. Time-resolved FRET experiments, however, generate a more complex data set, from which one can recover the distribution of distances (*Grinvald et al., 1972*; *Haas et al., 1975*; *Lakowicz et al., 1988*; *Lakowicz et al., 1990*). These distributions reveal the heterogeneity of each conformational state and distribution among different states. The distribution among two states at equilibrium allows for the determination of the free energy difference between the two states, a fundamental property of proteins that dictates their function.

Time-resolved FRET experiments complement single-molecule FRET and DEER spectroscopy experiments. Single-molecule FRET also measures the proportion of time the protein dwells in different states and, in addition, provides information on the transition rates between states, not currently possible with fluorescent lifetime measurements (*Mazal and Haran, 2019*; *Quast and Margeat, 2019*; *Metskas and Rhoades, 2020*; *Feng et al., 2021*; *Bartels et al., 2021*). Single-molecule measurements, however, are slow owing to technical limitations that generally require integration of photons for ≥10 ms. Therefore, any rearrangements faster than 10 ms will be lost. DEER is a powerful magnetic resonance-based method that reveals the detailed distance distributions between two spin labels attached to the protein (*Reginsson and Schiemann, 2011*; *Sahu and Lorigan, 2020*). To resolve distance distributions, however, DEER requires flash freezing the protein to capture the heterogeneity. Time-resolved FRET experiments are performed under physiological conditions and can distinguish any state that lasts longer than the lifetime of the fluorophore, 16 ns for Acd. In essence, time-resolved FRET experiments take a nanosecond-scale snapshot of the protein, allowing free energy measurements for even very fast transitions. DEER is also less sensitive than fluorescence, typically requiring low micromolar concentrations of label. In contrast, fluorescence measurements can be made with tens of nanomolar concentrations of fluorophore and can even be made within living cells.

The use of FRET to study distance distributions was first reported by Steinberg and coworkers using flexible polypeptides (*Haas et al., 1975*). It has subsequently been used in proteins (*Amir and Haas, 1986*), DNA (*Murchie et al., 1989*), ribozymes (*Walter et al., 1998*), and ribosomes (*Melcher et al., 2003*). With currently available instrumentation, distance distributions have been resolved, but it is not possible to distinguish the precise shape of the distribution. The determination of distance distributions directly from the lifetime data is an ill-posed problem, and the distance distributions have had to be represented by a model with a small number of parameters. The most commonly used parametric distribution model is a Gaussian (*Lakowicz et al., 1994a*; *Kulinski et al., 1997*), containing only two free parameters, the mean distance and standard deviation. Gaussian distributions are also commonly used as parametric models to fit DEER data (*Stein et al., 2015*). Whereas Gaussian distributions are most commonly used to fit fluorescence lifetimes, the distributions need not be Gaussian: other distance distributions such as Lorentzians have also been used (*Amir and Haas, 1986*; *Amir et al., 1992*; *Wu and Brand, 1994*).

We chose MBP as a model system because it has a well-characterized structure with well-understood conformational energetics. In addition to the >100 static MBP structures in the Protein Data Bank of clamshell open (apo) and closed (holo) states, the conformational dynamics of MBP—and related solute-binding proteins—have also been extensively studied. Previous DEER and double quantum coherence (DQC) EPR studies using small nitroxide spin labels have revealed distinct distance populations for apo and holo MBP with similar widths to those we observed here by time-resolved tmFRET (*Selmke et al., 2018*). Interestingly, previous NMR and DEER experiments have identified sparsely populated structural states of apo MBP with varying degrees of clamshell closure (*Tang et al., 2007*; *Selmke et al., 2018*; *Kaczmarski et al., 2020*). The existence of several conformations of apo MBP in equilibrium is likely responsible for the larger heterogeneity we observed for apo MBP compared to the maltose-bound state.

The approach presented here has some limitations that should be addressed in future work. Some of the parameters in the model are correlated, such as $\bar{r}$ and $\sigma$. The modeling would benefit from Bayesian statistical analysis to provide correlations and confidence intervals on the parameters and quantitative comparisons of different models, as has recently been done for DEER (*Sweger et al., 2020*). It is important to use narrower distributions to better resolve the backbone movements and the relative proportion of two conformational states. Even though we have used small probes and short linkers compared to most FRET studies, the heterogeneity in distances associated with each conformational state is likely still dominated by the rotameric states of the probes, particularly for $Cu^{2+}$-TETAC attached to a cysteine residue. The use of di-histidine metal-binding sites largely mitigates this problem, but bifunctional metal chelators might also be useful (*Beausang et al., 2012*). As demonstrated with MBP-295Acd, the lifetime of Acd may be affected by nearby residues or the local environment, a situation that can easily be identified by nonexponential lifetimes in the donor-only sample. This problem may be remedied by modeling the donor-only lifetimes with multiple exponentials (*Lakowicz et al., 1984*; *Lakowicz et al., 1987a*), mutating the offending residues, or alternative placement of the donor fluorophore. Also, because of its small size, Acd is a short wavelength fluorophore and is not very bright compared to larger fluorophores (*Lavis and Raines, 2008*; *Speight et al., 2013*). This decreases the sensitivity of the experiments and precludes fluorescence measurements from single molecules. In the future, brighter fluorophores and smaller, more rigid acceptors would improve the sensitivity and spatial resolution and make time-resolved tmFRET an even more faithful reporter of backbone distances and distributions. Even with these current limitations, however, the use of time-resolved tmFRET with Acd to resolve conformational distributions promises to provide many new insights into protein dynamics.

Applying our tmFRET approach with Acd to elucidate conformational equilibria in native environments will require overcoming significant challenges. We have previously shown that Acd can be incorporated into membrane proteins and soluble proteins in living mammalian cells (*Jones et al., 2021*). Endogenous fluorophores, such as NAD and NADH, contribute to a background problem when exciting Acd with 375 nm light, which can be partially overcome using 405 nm excitation. Because the lifetime of endogenous fluorophores is in the 2–3 ns range, we were able to use the much longer fluorescence lifetime of Acd to localize the Acd-incorporating proteins. In the case of tmFRET, in which the Acd lifetime would be reduced by the acceptor, it would be much more difficult to distinguish the Acd signal from autofluorescence. In addition, methods such as cell unroofing or permeabilizing cells would be required to gain solution access to intracellular donors for application of metal ion acceptors, and TETAC would likely be rendered unreactive in a reducing cellular environment. Until these problems can be overcome, tmFRET using Acd as a donor will likely be limited to in vitro applications. Even with these current limitations, however, the use of time-resolved tmFRET with Acd to resolve conformational distributions promises to provide many new insights into protein energetics and dynamics.

## Materials and methods

**Key resources table**

| Reagent type (species) or resource | Designation | Source or reference | Identifiers | Additional information |
|---|---|---|---|---|
| Cell line (*Homo sapiens*) | HEK293T/17 | ATCC | ATCC: CRL-11268; RRID:CVCL_1926 | |
| Recombinant DNA reagent | NESAcdRS82.pUC57 | BioBasic | | |

*Continued on next page*

*Continued*

| Reagent type (species) or resource | Designation | Source or reference | Identifiers | Additional information |
|---|---|---|---|---|
| Recombinant DNA reagent | pAcBac1.tR4-MbPyl | Plasmid # 50832; http://n2t.net/addgene:50832; | RRID:Addgene_50832 | |
| Recombinant DNA reagent | pANAP | Addgene: DOI: 10.1021/ja4059553 | Addgene: 48696 | |
| Recombinant DNA reagent | DN-eRF1 (peRF1-E55D.pcDNA5-FRT) | Jason Chin: DOI: 10.1021/ja5069728 | | |
| Recombinant DNA reagent | FLAG-MBP1-K295TAG-W340A.pcDNA3-k | Addgene | Plasmid # 126627 | |
| Recombinant DNA reagent | FLAG-MBP1-K295TAG-W340A-S233H-T237C.pcDNA3-k | Addgene | Plasmid # 127402 | |
| Recombinant DNA reagent | FLAG-MBP1-E322TAG-W340A.pcDNA3-k | Addgene | Plasmid # 127404 | |
| Recombinant DNA reagent | FLAG-MBP1-E322TAG-E309C-W340A.pcDNA3-k | Addgene | Plasmid # 127408 | |

## Constructs, cell culture, and transfection

All target constructs were made in the pcDNA3.1 mammalian expression vector (Invitrogen, Carlsbad, CA), except as noted below. The MBP constructs with W340A mutation, pAnap, and DN-eRF1 were previously described (*Schmied et al., 2014*; *Gordon et al., 2018*). The amber stop codons (TAG) in MBP were introduced at positions 295 (MBP-295TAG) and 322 (MBP-322TAG). For each stop codon, three constructs were produced: no metal-binding site (i.e., wild-type), a di-histidine, and a cysteine. For MBP-295TAG, the di-histidines were introduced at positions 233 and 237 (MBP-295TAG-Y307F-HH), and the cysteine was introduced at position 237 (MBP-295TAG-C and MBP-295TAG-Y307F-C). MBP-295TAG-C and MBP-295TAG-Y307F-C also contained the 233H mutation. For MBP-322TAG, the di-histidines were introduced at positions 305 and 309 (MBP-322TAG-HH), and the cysteine at position 309 (MBP-322TAG-C). The aminoacyl tRNA synthetase for Acd with a nuclear export sequence, RS82, was synthesized by BioBasic and subcloned into pAcBac1.tR4-MbPyl (Addgene, Cambridge, MA) (pAcBac1.tR4-AcdRS82) for transient transfection (*Jones et al., 2021*).

HEK293T/17 cells were obtained from ATCC (Manassas, VA; #CRL-11268; RRID:CVCL_1926) and were expanded only once immediately upon receipt. Cells were used for no more than 35 passages and replenished from frozen aliquots of the originally expanded stock. Unused samples of the originally expanded stock were used for no more than 3 years, at which point a new stock was purchased. No additional authentication was performed. Testing for mycoplasma contamination was performed using the MicroFluor Mycoplasma Detection kit (catalog #M7006; Thermo Fisher Scientific, Waltham, MA) at the time of the last passage and no contamination was detected.

HEK293T/17 cells were plated in six-well trays on glass coverslips. DMEM (Life Technologies, # 11995-065) supplemented with 10% fetal bovine serum and penicillin-streptomycin (50 U/mL) was used for HEK293T/17 culture. Cells were split twice a week and never exceeded 95% confluency. Cells were transfected at ≈25% confluency with a total of 1.6 µg of DNA and 10 µL of Lipofectamine 2000 (Invitrogen) per well. The 1.6 µg of DNA consisted of 0.9 µg of target gene, 0.3 µg of pANAP or pAcBac1.tR4-AcdRS82, and 0.4 µg of DN-eRF1. The DNA/Lipofectamine mix was prepared in 300 µL Opti-MEM (Invitrogen) per well. For transfection, cells were incubated in growth medium without antibiotics for 4–6 hr at 37 °C with 5% $CO_2$. After incubation, the medium was replaced with one including antibiotics and supplemented with either 20 µM L-Anap-ME (AsisChem, Waltham, MA) or 450 µM Acd. The L-Anap-ME was made as a 10 mM stock in ethanol and stored at –20 °C. Acd was synthesized as previously described (*Speight et al., 2013*) and made as a 30 mM stock in water with drop-wise addition of 1 M NaOH to solubilize Acd and stored at 4 °C. Trays were wrapped with aluminum foil to block light and incubated at 37 °C until use. Cells were harvested approximately 48 hr after transfection unless otherwise specified. Cells were washed twice with PBS, and cell pellets from nine wells of six-well trays were stored at –20 °C until use.

## Protein purification, western blot analysis, and fluorescence size-exclusion chromatography

For fluorometry experiments, between 1 and 4 frozen pellets were thawed and resuspended in 0.8 mL Potassium Buffer Tris (KBT) (in mM: KCl 130; Trizma Base 30; pH 7.4) supplemented with cOmplete mini EDTA-free protease inhibitor cocktail (Sigma-Aldrich, St. Louis, MO). The suspension was sonicated using a Sonifier 450 with MicroTip (Branson, Danbury, CT) with settings of power = 4 and duty cycle = 50% for a total of 10 pulses. Lysed cells were then spun in a benchtop refrigerated centrifuge at 13,000 rpm for 20–30 min at 4 °C, and the cleared lysate was moved to a new tube.

Anti-FLAG M2 affinity gel (Sigma-Aldrich) was prepared by rinsing 50–100 µL of slurry per sample five times with 1 mL KBT. Cleared lysate was then added to the rinsed gel and nutated at 4 °C for 1 hr. Tubes were wrapped in aluminum foil to prevent photodamage. The gel was then rinsed five times with 1 mL of KBT; each rinse cycle consisted of a 1 min incubation period in KBT followed by a nutation period of about 40 s in order to ensure that the gel was free of unincorporated L-Acd. Between 0.25 and 0.5 mL of a 100–200 ng/mL solution of FLAG peptide (Sigma-Aldrich) in KBT was added to the rinsed beads, which were then nutated at 4 °C for 1 hr to elute the protein. The purified protein was harvested by spinning down the beads and collecting the supernatant. The protein was stored at –80 °C.

For western blot analysis, we used 25 µL of lysis buffer per 10 mg cells. Lysis buffer included 25 mM triethanolamine (pH 8), 130 mM NaCl, 8 mM digotonin, 20% l glycerol, and 1 % Halt protease inhibitor cocktail (Thermo Fisher Scientific). Cleared cell lysates were run on NuPage 4–12% bis-tris gels (Thermo Fisher Scientific) with MES SDS running buffer (50 mM MES, 50 mM Tris, 0.1% SDS, and 1 mM EDTA). Proteins were transferred to PVDF membranes using a BioRad Transblot SD (Hercules, CA) transfer cell with Bjerrum/Schafer-Nielsen transfer buffer with SDS (*Damm, 1986*). Membranes were blocked in 5% Milkman Instant Low Fat Dry Milk (Marron Foods, Harrison, NY) in TBS-T (20 mM Tris, 137 mM NaCl, 0.1% Tween20, pH 7.6) for either 1 hr at room temperature or overnight at 4 °C. Anti-FLAG primary antibody (Cat F3165; Sigma-Aldrich) was used at a dilution of 1:20,000 and secondary antibody HRP-linked anti-mouse IgG (Cat NA931; Amersham, Pittsburgh, PA) was used at 1:30,000 dilution in TBS-T. Secondary antibodies were visualized with Super Signal West Femto Substrate (Thermo Fisher) and imaged with a Proteinsimple gel imager (San Jose, CA).

Gel filtration was performed on a Shimadzu Prominence HPLC (Kyoto, Japan) with a GE Superdex 200 Increase 5/150 GL column run at 0.3 mL/min with KBT as the running buffer. Tryptophan absorption was measured at 280 nm and Acd was excited at 385 nm with fluorescence recorded at 450 nm.

## Fluorometry and spectrophotometry

Starna (Atascadero, CA) sub-micro-fluorometer cells (100 µL) were used for both fluorometry and spectrophotometry. Absorption measurements were made using a Beckman Coulter DU 800 spectrophotometer (Brea, CA). Fluorometry experiments were performed using a Jobin Yvon Horiba FluoroMax-3 spectrofluorometer (Edison, NJ). For emission spectra of Acd, we used excitation wavelengths of 370–385 nm, as indicated in the figure legends, and for Anap, we used excitation wavelengths of 350–370 nm, as indicated in the figure legends. We used 5 nm slits for excitation and emission, except for experiments to measure quantum yield in which we used 1 nm slits and photobleaching experiments, which used 14.1 nm excitation slits. For Acd time-course measurements, we excited samples at 385 nm and recorded the emission at 425 nm at alternating 10 and 30 s intervals using the anti-photobleaching mode of the instrument. For Anap time-course measurements, we excited samples at 350 nm and recorded the emission at 480 nm. Reagents ($Cu^{2+}$, $Cu^{2+}$-TETAC, EDTA, and TCEP) were added manually as 50– 200× stocks during the period between measurements by pipetting up and down in the cuvette without removing it from the instrument. To minimize the loss of fluorescence signal due to the adhesion of protein to the interiors of the cuvette and pipette tip, both were passivated with 2% bovine serum albumin (Sigma-Aldrich) in KBT, followed by thorough rinsing with ultrapure water before use.

Protein samples were diluted 1:50 to 1:100 in KBT to keep the fluorescence intensity within the linear range of the fluorometer. $Cu^{2+}$ was prepared from $CuSO_4$ as a 110 mM stocks in water, then diluted to make stocks of lower concentrations. TETAC (Toronto Research Chemicals, Toronto, Canada) was prepared as a 100 mM stock in DMSO and stored at –20 °C until the day of use. To prepare $Cu^{2+}$-TETAC, equal volumes of 100 mM TETAC stock and 110 mM $CuSO_4$ stock were mixed

together and allowed to incubate for 5 min at room temperature as the solution turned a darker shade of blue, indicating coordination of $Cu^{2+}$ by the cyclen ring. This mixture was then diluted with water to concentrations of 1.1 mM $Cu^{2+}$ and 1 mM TETAC. Incubation at high concentrations and the 10% excess of $Cu^{2+}$ ensured that all of the TETAC was bound with $Cu^{2+}$. This stock solution was then diluted 1:100 when added to the cuvette for fluorometer experiments, giving a final concentration of 10 μM $Cu^{2+}$-TETAC. 0.5 M Bond-Breaker tris(2-carboxyethyl)phosphine (TCEP) stock solution (Thermo Fisher) was diluted 1:200 in the cuvette for a final concentration of 2.5 mM. EDTA (Sigma-Aldrich) was made as a 0.5 M stock in water, with the pH adjusted to pH 7 using HCl. This 0.5 M solution of EDTA in water was then diluted 1:50 in the cuvette for a final concentration of 10 mM.

## Calculation of quantum yield

Quantum yields of Anap and Acd in KBT were determined relative to quinine in 0.5 M $H_2SO_4$ using a quantum yield of 0.546 measured with 366 nm excitation for the quinine standard (*Brouwer, 2011*) using the following equation (*Lakowicz, 2006*):

$$Q_M = Q_{quinine} \frac{slope_M}{slope_{quinine}} \frac{\eta_M^2}{\eta_{quinine}^2}$$

(1)

where $Q$ is the quantum yield, slope refers to the slope of the linear fits to the data, and $\eta$ is the refractive index for the Acd or Anap (M) and quinine samples. This method gives a quantum yield value for Acd of 0.8 and a quantum yield value for Anap of 0.32.

## Measurement of FRET efficiency from steady-state fluorescence intensity

For each time-course experiment in the fluorometer, the background signal in the absence of protein was first subtracted from the protein-containing signal. The fraction of fluorescence unquenched (*F*) was defined as follows:

$$F = \frac{Fluoresence_{metal}}{Fluorescence_{no\ metal}}$$

(2)

To determine the FRET efficiency, *E*, we corrected for nonspecific decreases in fluorescence (e.g., bleaching or loss of protein) with constructs without a cysteine or di-histidine motif as previously described (*Gordon et al., 2018*) using the following equations:

$$E = 1 - \frac{F_{Cys}}{F_{no\ Cys}} \text{ and } E = 1 - \frac{F_{HH}}{F_{no\ HH}}$$

(3)

where *Cys*, *no Cys*, *HH*, and *no HH* refer to the presence or absence of a cysteine or di-histidine motif, respectively, in the construct. To calculate the mean and standard error of the mean for *E*, we used the mean and standard error of the mean for our *F* measurements (i.e., $F_{HH}$, $F_{no\ HH}$, $F_{Cys}$, and $F_{no\ Cys}$) in Monte Carlo resampling ($1 \times 10^6$ cycles; NIST Uncertainty Machine v1.3.4; *Lafarge and Possolo, 2016*).

## Determination of $R_0$ and the fraction of donor only ($f_D$)

$R_0$ values for FRET between Acd and either $Cu^{2+}$-di-histidine or $Cu^{2+}$-TETAC were calculated using the measured donor emission spectrum and quantum yield and the measured absorption spectrum for each type of bound metal using the following equation (*Lakowicz, 2006*):

$$R_0 = C\sqrt[6]{JQ\eta^{-4}\kappa^2}$$

(4)

where *C* is a scaling factor, *J* is the normalized spectral overlap of the emission of the donor and absorption of the acceptor, *Q* is the quantum yield of Acd (see above), $\eta$ is the index of refraction (1.33 in our case), and $\kappa^2$ is the orientation factor, assumed to be 2/3. This assumption is justified for at least three reasons. (1) The metal ion acceptor has multiple transition dipole moments. In the case of a mixed polarization donor-acceptor pair, in which one probe is freely rotating and one is immobile, $1/3 < \kappa^2 < 4/3$ (*Haas et al., 1978*). Thus, even if our donor were immobile, an assumption of $\kappa^2 = 2/3$ gives a maximum error in $R_0$ of about ±11% . (2) Our data reveal a distribution of donor-acceptor distances, introducing additional sources of randomized orientation and further reducing the potential error due

to the $\kappa^2 = 2/3$ assumption. (3) Using $\kappa^2 = 2/3$ for our $R_0$ calculations in this and other tmFRET studies gives the expected distances based on known structures. We based the determination of $R_0$ on the emission spectrum of the free amino acid because of the more optimal signal-to-noise above 490 nm where the overlap with the acceptor absorption spectra occurs, and the indistinguishable shape of emission spectra of Acd-incorporating proteins and Acd in this region of overlap (**Figure 8—figure supplement 1**).

## Measurement of frequency-domain fluorescence lifetime using FLIM

The theory underlying our FRET measurements with fluorescence lifetimes is well described elsewhere (**Lakowicz, 2006**). Briefly, FRET decreases the fluorescence lifetime of a donor fluorophore by providing an additional path by which an excited state electron can lose its energy. When using a pulsed excitation source and measuring fluorescence in the time domain, the decrease in lifetime is readily apparent as a faster decay in fluorescence intensity after excitation (**Figure 8—figure supplement 2A**). When using a frequency ($\omega$)-modulated excitation source, the lifetimes of donor in the absence and presence of acceptor are determined from the phase delays ($\varphi_\omega$) and modulation ratios ($m_\omega$) at each frequency (**Figure 8—figure supplement 2B**). With our frequency-domain instrument, the frequency dependence of both $\varphi_\omega$ and m$_\omega$ (**Figure 8—figure supplement 2C**) is required to resolve complex lifetimes. A similar analysis can be performed using a time-domain instrument (**Lakowicz, 2006**).

Frequency-domain fluorescence lifetime data were collected using a Q2 laser scanner and A320 FastFLIM system (ISS, Inc, Champaign, IL) mounted on a Nikon TE2000U microscope (Melville, NY) and VistaVision software (ISS, Inc). Acd, Anap, or Atto 425 (the standard for calibration of the fluorescence lifetime) were excited using a 375 nm pulsed diode laser (ISS, Inc), driven by FastFLIM at the repetition rate of 10 MHz, with a 387 nm long-pass dichroic mirror, and emission was collected using a 451/106 nm band-pass emission filter and Hamamatsu model H7422P PMT detector. Affinity purified protein was used full strength in KBT buffer. For each experiment, 10–11 µL of fluorescent sample was pipetted onto an ethanol-cleaned #1.5 glass coverslip mounted directly above the 60 × 1.2 NA water-immersion objective. Other reagents (maltose, Cu$^{2+}$-TETAC, Cu$^{2+}$, TCEP, or EDTA) were pipetted directly into the sample drop and mixed at the final concentrations indicated in the text. For each condition, 256 × 256 confocal images were collected with a pinhole of 200 µm and a pixel dwell time of 1 ms. The pixels were averaged together for analysis, except as described for the phasor plot.

The experimental phase delays ($\varphi_\omega$) and the modulation ratios ($m_\omega$) of the fluorescence signal in response to an oscillatory stimulus with frequency $\omega$ were obtained using VistaVision software from the sine and cosine Fourier transform of the phase histogram $H(p)$, subject to the instrument response function (IRF) calibrated with 2 µM Atto 425 in water with a lifetime of 3.6 ns (**Lakowicz, 2006**; **Colyer et al., 2008**; **Digman et al., 2008**).

## Analysis of frequency-domain fluorescence lifetime data

The theoretical estimates for $\varphi_\omega$ and $m_\omega$ were calculated from a model for fluorescence lifetime and FRET that assumes a single- or double-exponential donor fluorescence lifetime with one or two Gaussian-distributed distances between the donor and acceptor as previously described with some modification (**Cheung et al., 1991**; **Lakowicz et al., 1991**; **Lakowicz et al., 1994b**; **Lakowicz, 2006**). The phase delays ($\varphi_\omega$) and modulation ratios ($m_\omega$) were calculated as a function of the modulation frequency ($\omega$) using the following equations:

$$\varphi_\omega = \arctan\left(\frac{N_\omega}{D_\omega}\right) + \omega t_0 \qquad (5)$$

$$m_\omega = \sqrt{N_\omega^2 + D_\omega^2} \qquad (6)$$

where $N_\omega$ corresponds to the imaginary component and $D_\omega$ corresponds to the real component of fluorescence and $t_0$ is the time shift of the IRF (**Figure 8E**). The imaginary and real components for the fluorescence response with a contaminating background fluorescence were calculated using the following equations:

$$N_\omega = (1 - f_B) \frac{1}{J} \int_0^\infty \sum_{i=1}^{2} \frac{P(r)\,\alpha_{D_i}\omega\tau_{DA_i}^2}{1 + \omega^2\tau_{DA_i}^2} dr + f_B m_{\omega B} \sin(\varphi_{\omega B}) \tag{7}$$

$$D_\omega = (1 - f_B) \frac{1}{J} \int_0^\infty \sum_{i=1}^{2} \frac{P(r)\,\alpha_{D_i}\tau_{DA_i}}{1 + \omega^2\tau_{DA_i}^2} dr + f_B m_{\omega B} \cos(\varphi_{\omega B}) \tag{8}$$

where $f_B$ is the fraction of the fluorescence intensity due to background (**Figure 8D**), and $\varphi_{\omega B}$ and $m_{\omega B}$ are the phase delay and modulation ratio of the background fluorescence measured from samples of KBT without or with maltose. $\alpha_i$ is the amplitude of the $i$th component of the donor-only fluorescence decay, and $\alpha_1 + \alpha_2 = 1$ (**Figure 8A**). The normalization factor $J$ is given by

$$J = \int_0^\infty \sum_{i=1}^{2} P(r)\,\alpha_{D_i}\tau_{DA_i}dr \tag{9}$$

The decay time constant of the $i$th component of the donor lifetime in the presence of acceptor ($\tau_{DAi}$) is given by

$$\frac{1}{\tau_{DA_i}} = \frac{1}{\tau_{D_i}} + \frac{1}{\tau_{D_i}}\left(\frac{R_0}{r}\right)^6 \tag{10}$$

where $\tau_{D_i}$ is the decay time constant of the donor in the absence of acceptor (**Figure 8A**), $r$ is the distance between the donor and acceptor, and $R_0$ is the characteristic distance for the donor-acceptor pair (**Figure 8B**).

The apparent FRET efficiency based on the donor quenching of the acceptor was calculated using the following equation:

$$E = 1 - J/\sum_{i=1}^{2} \alpha_{D_i}\tau_{DA_i} \tag{11}$$

The distribution of donor-acceptor distances $P(r)$ was assumed to be the sum of up to two Gaussians:

$$P(r) = (1 - f_D) \sum_{i=1}^{2} \frac{A_i}{\sigma_i\sqrt{2\pi}} exp\left[-\frac{1}{2}\left(\frac{r - \bar{r}_i}{\sigma_i}\right)^2\right] + \frac{f_D}{\sigma_i\sqrt{2\pi}} exp\left[-\frac{1}{2}\left(\frac{r - 150\,\text{Å}}{0.1\,\text{Å}}\right)^2\right] \tag{12}$$

where $A_i$, $\bar{r}_i$, and $\sigma_i$ are the amplitude, mean, and standard deviation of the $i$th Gaussian, respectively, and $A_1 + A_2 = 1$ (**Figure 8C**). The fraction of donor only ($f_D$) was modeled as a narrow Gaussian with a mean distance of 150 Å and a standard deviation of 0.1 Å, too far to exhibit any detectable FRET.

The phase delay and modulation ratios displayed were corrected for the background fluorescence and time shift in the IRF using the following equations:

$$\varphi_\omega^{corr.} = \arctan\left(\frac{m_\omega \sin(\varphi_\omega - \omega t_0) - f_B m_{\omega B} \sin(\varphi_{\omega B})}{m_\omega \cos(\varphi_\omega - \omega t_0) - f_B m_{\omega B} \cos(\varphi_{\omega B})}\right) \tag{13}$$

$$m_\omega^{corr.} = \sqrt{\left[\frac{m_\omega \sin(\varphi_\omega - \omega t_0) - f_B m_{\omega B}\sin(\varphi_{\omega B})}{1 - f_B}\right]^2 + \left[\frac{m_\omega \cos(\varphi_\omega - \omega t_0) - f_B m_{\omega B}\cos(\varphi_{\omega B})}{1 - f_B}\right]^2} \tag{14}$$

This model for fluorescence lifetimes and FRET was implemented in Igor (Wavemetrics, Lake Oswego, OR; code available at https://github.com/zagotta/FDlifetime_program; **Zagotta, 2021**). The model was simultaneously fit to the phase delay, modulation ratio, and steady-state quenching data with $\chi^2$ minimization. There are 12 parameters in the parameter vector ($f_D$, $\tau_{D1}$, $\alpha_1$, $\tau_{D2}$, $R_0$, $\bar{r}_1$, $\sigma_1$, $A_2$, $\bar{r}_2$, $\sigma_2$, $t_0$, $f_B$) (**Figure 8**, blue variables). Since our donor fluorophore was determined to have a single exponential decay lifetime in our MBP constructs, the donor-only data were fit with three free parameters ($\tau_{D1}$, $t_0$, $f_B$) where $f_D$=1 and $\alpha_1$=1. These $\tau_{D1}$ and $\alpha_1$ values were then used for the remainder of the experiments with that sample.

For experiments using $Cu^{2+}$-TETAC as an acceptor, the donor-acceptor data at zero maltose concentration were assumed to arise from donor-acceptor distances with a single Gaussian distribution, plus an 8% contribution from donor only. This estimate of donor-only sites that were not labeled by $Cu^{2+}$-TETAC was made using a cysteine-reactive quencher with a long $R_0$ value (38 Å; Tide

Quencher 1 maleimide; AAT Biosciences, Sunnyvale, CA; *Figure 3—figure supplement 2*). When labeling with 100 µM $Cu^{2+}$, labeling efficiency was assumed to be 100% (*Gordon et al., 2018*). For these experiments, four parameters were allowed to vary $(\bar{r}_1, \sigma_1, t_0, f_B)$, where $f_D$ and $R_0$ were previously determined from previous spectroscopy experiments, and $A_2 = 0$. Similarly, the donor-acceptor data at saturating maltose concentration were assumed to arise from donor-acceptor distances with a single Gaussian distribution, plus a small amount of donor only, and four parameters were allowed to vary $(\bar{r}_2, \sigma_2, t_0, f_B)$ where $A_2 = 1$. The donor-acceptor data at subsaturating maltose concentration were fit with three parameters $(A_2, t_0, f_B)$ with all other parameters the same as the preceding donor-acceptor experiments. Finally, the data for reversal of acceptor binding using TCEP or EDTA were fit with three free parameters $(f_D, t_0, f_B)$ with all other parameters the same as the preceding experiments.

## In silico distance modeling

Rotamer distributions for donor and acceptor sidechains were modeled from the published crystal structures of apo (PDB 1omp) and maltose-bound (PDB 1anf) MBP using the program mtsslWizard (*Hagelueken et al., 2012*). Starting models for Acd, $Cu^{2+}$-TETAC, and $Cu^{2+}$-histidine were built in Avogadro (version 1.2.0, http://avogadro.cc) and geometry optimized with MMFF94 (Acd) or UFF ($Cu^{2+}$-histidine and $Cu^{2+}$-TETAC) force fields. For the $Cu^{2+}$-histidine structure, copper was bound to N $\varepsilon$ 2 of the histidine imidazole at a fixed bond length of 2.03 Å to approximate square planar coordination geometry. Starting structures were imported as custom amino acids into the mtsslWizard PyMOL plugin. For Acd and $Cu^{2+}$-TETAC, 200 rotamers were generated at each site using the 'tight' van der Waals (vdW) constraint setting (3.4 Å cutoff, no clashes allowed). For $Cu^{2+}$-di-histidine, 200 $Cu^{2+}$-histidine rotamers were first generated at each residue of the di-histidine motif (233 and 237) using 'loose' vdW constraints (2.5 Å cutoff, 5 clashes allowed). Next, rotamers lacking a companion in the $i \pm 4$ site with a $Cu^{2+}$-$Cu^{2+}$ distance of 1 Å or less were excluded. Distance distributions were obtained by normalization of the donor-acceptor distance histograms with all rotamers included, with distances calculated between the $Cu^{2+}$ ion of $Cu^{2+}$-TETAC/$Cu^{2+}$-di-histidine and the carbonyl carbon of the acridone ring of Acd.

## Acknowledgements

The research reported in this publication was supported by the National Eye Institute of the National Institutes of Health under award numbers R01EY010329 (to WNZ), the National Institute of General Medical Sciences of the National Institutes of Health under award numbers R01GM125351 (to SEG) and R01GM127325 (to WNZ), the National Science Foundation under award number CHE-1708759 (to EJP), and the following additional awards from the National Institutes of Health: S10RR025429, P30DK017047, R01GM125351-13S, and P30EY001730. CMJ thanks the National Institutes of Health for funding through the Structural Biology and Molecular Biophysics Training Program (T32 GM-008275). We thank Mika Munari for expert technical assistance and Jacob LW Morgan for helpful discussions. The authors declare no competing financial interests.

## Additional information

### Funding

| Funder | Grant reference number | Author |
| --- | --- | --- |
| National Eye Institute | R01EY010329 | William N Zagotta |
| National Institute of General Medical Sciences | R01GM125351 | Sharona E Gordon |
| National Institute of General Medical Sciences | R01GM127325 | William N Zagotta |
| National Science Foundation | CHE-1708759 | E James Petersson |
| National Institutes of Health | S10RR025429 | Sharona E Gordon |

| Funder | Grant reference number | Author |
|---|---|---|
| National Institute of Diabetes and Digestive and Kidney Diseases | P30DK017047 | Sharona E Gordon |
| National Institute of General Medical Sciences | R01GM125351-13S | Sharona E Gordon |
| National Eye Institute | P30EY001730 | William N Zagotta Sharona E Gordon |
| National Institute of General Medical Sciences | T32 GM-008275 | Chloe M Jones |
| National Institute of General Medical Sciences | R01GM131168 | Ryan A Mehl |

The funders had no role in study design, data collection and interpretation, or the decision to submit the work for publication.

### Author contributions

William N Zagotta, Sharona E Gordon, Conceptualization, Data curation, Formal analysis, Funding acquisition, Investigation, Methodology, Project administration, Resources, Software, Supervision, Validation, Visualization, Writing - original draft, Writing – review and editing; Brandon S Sim, Anthony K Nhim, Marium M Raza, Yarra Venkatesh, Chloe M Jones, Data curation, Formal analysis, Investigation, Methodology, Writing – review and editing; Eric GB Evans, Conceptualization, Data curation, Formal analysis, Investigation, Methodology, Software, Visualization, Writing – review and editing; Ryan A Mehl, Conceptualization, Data curation, Methodology, Writing – review and editing; E James Petersson, Conceptualization, Data curation, Funding acquisition, Methodology, Writing – review and editing

### Author ORCIDs

William N Zagotta http://orcid.org/0000-0002-7631-8168
Yarra Venkatesh http://orcid.org/0000-0002-4478-1553
Ryan A Mehl http://orcid.org/0000-0003-2932-4941
E James Petersson http://orcid.org/0000-0003-3854-9210
Sharona E Gordon http://orcid.org/0000-0002-0914-3361

### Decision letter and Author response

Decision letter https://doi.org/10.7554/eLife.70236.sa1
Author response https://doi.org/10.7554/eLife.70236.sa2

## Additional files

### Supplementary files

• Transparent reporting form

### Data availability

Source data files have been included with the submission. Code used for data analysis has been posted to https://github.com/zagotta/FDlifetime_program (copy archived at https://archive.software-heritage.org/swh:1:rev:de8978e598f420cc7c19bde8bbb02f78d6cf41fd).

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
