## [Decision Letter]

**Acceptance summary:**

The authors of this work combine powerful fluorescence spectroscopy approaches with a new fluorescent non-canonical amino acid to develop a work flow that promises to provide detailed structural information and protein energetics at the same time. These methods include improved incorporation of a long-lifetime fluorescent non-canonical amino acid in an eukaryotic expression system in combination with transition metal fluorescence resonance energy transfer to obtain fluorescence signals that can be parsed into the distinct conformations of the protein being studied. The authors have answered the comments of reviewers and this improved version of the manuscript should be of interest to biochemists, biophysicists and structural biologists.

**Decision letter after peer review:**

Thank you for submitting your article "An improved fluorescent noncanonical amino acid for measuring conformational distributions using time-resolved transition metal ion FRET" for consideration by *eLife*. Your article has been reviewed by 3 peer reviewers, including Leon D Islas as the Reviewing Editor and Reviewer #2, and overseen by Kenton Swartz as the Senior Editor.

Essential Revisions:

The reviewers judge that the use of the non-canonical amino acid (NCA) Acd is an significant advance over ANAP and other fluorescent NCAs. The authors present a strong case for these advantages using the purified MBP as a protein model. However, the reviewers argue that MBP is a rare case of a well characterized and behaved soluble protein and that for ACd and the methods used to really be an improvement over available approaches, the authors should demonstrate its use in a membrane protein or at least a membrane-based system.

*Reviewer #1:*Zagotta et al., report a new noncanonical amino acid L‐Acridonylalanine (Acd) as a tmFRET donor for measuring distances in proteins. The authors first characterize Acd from various aspects, and prove its better performance than the previously reported Anap, including high quantum yield, high stability in the cell environment, and long, single-exponential lifetime. By using maltose binding protein (MBP) expressed in mammalian cells as a model system, the authors further demonstrate that Acd can be used to identify multiple protein conformations at subsaturating maltose concentrations in time-resolved tmFRET. This work provides a tool for detecting conformational heterogeneity and structural dynamics. In large the paper described an improvement compared to a previous amino acid, ANAP. I think the improvements are of interest for a biophysical audience, even if not groundbreaking.

I feel that UV or 405 excitable probes as well as copper based probes have limited applicability in cell biology. MBP expressed in cells is also one of the most-well behaving model system one can think of.

The following aspects of data analysis need to be clarified and extended:

The authors claimed that for experiments using Cu2+-TETAC as an acceptor, 8% contribution was from donor-only. However, how such a fraction of donor-only was estimated is missing in the text. The authors should clarify. Furthermore, for the experiments using di-histidine as an acceptor, the authors didn't provide enough information to show that the donor-only population was negligible. In Figure 7D, the center of mass of the measured data for 10 mM maltose (magenta) on the phasor plot seems to shift a bit towards the direction of donor-only population compared with the center of f_model, which may indicate the existence of donor-only population. If the donor-only fraction is not negligible, then the tmFRET results in Figure 7 should be fitted with 3-component (donor-only, Apo, and holo), rather than 2-component model.

The donor is still a dipole, and the Foerster equation only holds when sufficient rotational mobility exists. This is why FRET usually uses the long linkers, which they stress as a disadvantage. Maybe the authors can comments on this. I believe their long linker statement might only apply to their acceptor, but for their donor it is actually a concern. Polarization effects would thus need to be discussed and accounted for.

The authors claimed that Anap, but not Acd, appeared to undergo a site-specific chemical change in cells. Yet, a small but noticeable shift can be seen for MBP-295Acd in in Figure 1 supp. 1A. Can the authors give some comments on that? Given the results that mutation of Y307 tyrosine to F can prevent the quenching of MBP-295Acd, one may attribute the shift of MBP-295Anap or MPB-295Acd in Figure 1 supp 1A,B to the slow energy transferring or blue-shifting by Y307 during cell culture. Since Acd is quenched more slowly than Anap, the shift may show not as much as Anap. In such sense, is it better to harvest cells 24hr after transfection rather than 48hr if one wants to use Acd to measure other kinds of proteins without site-mutations?

The authors mentioned that R0 value calculated for the Acd/Cu2+-TETAC FRET pair (14.9 Á…) is lower than for the Anap/Cu2+-TETAC FRET pair (17.2 Á…). Did the authors measure the R0 for all the mutants, e.g. MBP-295-C, MBP-295-307F, and MBP-322-C. Do they show same R0 or not for same noncanonical amino acid? What is the R0 for Acd/Cu2+-di-histidine FRET pair? The author should provide a list for that.

*Reviewer #2:*

Understanding of protein conformational dynamics is often achieved deploying special purpose methods. Likewise, determination of protein structures requires involved specialized methodology. However, protein structure and function go hand in hand. Achieving a picture of protein dynamics based on the static structures of X-ray crystallography and Cryo-EM methods is challenging at best and impossible in practice.

The authors of this manuscript have combined some of the most promising experimental approaches in fluorescence spectroscopy to develop a work flow and materials that promise to provide relatively high resolution structural information and protein dynamics at the same time. The authors have improved incorporation of a fluorescent non-canonical amino acid in an eukaryotic expression system and show that this is incorporated with specificity and efficiency. They proceed to utilize this fluorescent probe with transition metal FRET to obtain fluorescence signals that can be parsed into the distinct conformations of the protein that produce them. This allows the method to estimate the occupation probability of states at equilibrium which can then be translated into free energy information.

The authors have produced a clear manuscript that explains the usefulness of the method. I specially like they have selected a well characterized protein such as Maltose Binding Protein (MBP) to display the power of their approach.

While the authors present their results as a way to obtain dynamic information; it should be pointed out that as it stands the method can estimate equilibrium constants but not rate constants, which are needed to say that one has access to the dynamics.

This is a significant advance that should find broad application in the study of a wide range of proteins in different physiological scenarios.

The manuscript is very well written and clear.

– I think the presentation will benefit from a figure showing the raw fluorescence data and explaining how the phase and amplitude are obtained and interpreted.

– While the authors suggest a general applicability of the method, they have only show its application to a small, well behaved soluble protein. Have the authors applied it, even if preliminarily, to membrane proteins or other, more complex proteins?

– The distribution of distances is assumed to be Gaussian, please provide a justification for this assumption. Is it a necessary part of the method? What happens when the distribution of distances is skewed?

– While Acd is presented as an improvement over ANAP, the differences in photo physic properties really not that great, except the lifetime. Did you try to carry out similar lifetime measurements with ANAP? It seems the resolution of the instrumentation should be sufficient.

*Reviewer #3:*

In this paper by Zagotta and Gordon a new updated form of transition metal ion FRET-based structure mapping is presented. Here, the authors specifically present an ultra-small fluorescent non-canonical amino acid Acd that has useful photo-physical properties. They genetically incorporate this fluorophore into specific locations in maltose-binding protein along with cysteine-modified metal chelators or introduced histidine/metal binding sites and map distances and distance changes that result from ligand biding. The key advancement is the very long and single exponential nature of the fluorescence lifetime decay of Acd. This property allows for more accurate and easier fitting of the lifetimes for the determination of not only distances, but the range and populations of distances, within the population of proteins in solution. This paper is excellent. It is technically careful and creative. It moves the field towards understanding not only how proteins are structured, but how they are dynamically structured. The methods introduced in this paper, while challenging, can be applied to many different protein systems and the analysis framework is likewise widely applicable. The distances, dynamics, and energetics obtained from this work are incredible. I find the ~sub-2 Å matching between the expected backbone distances and the fine-scale dynamics particularly exciting. The move towards very small FRET probes with limited excursions, and a generalized framework for analysis, makes this work useful for the field of structural biology.

Recommendations to the authors to improve this manuscript and additional questions.

1. Understanding frequency domain plots can be tricky for the non-specialist. Standard lifetime plots in the time domain are generally (a bit) more intuitive and common. The authors do a good job of addressing the plots of lifetime in the frequency domain but I would suggest a more robust description of these plots and their expected changes in the main text to aid the reader in understanding these data. This is particularly important because this method will be generally useful for a broad audience. For example, why are both the modulation ration and phase delay necessary to plot on the graphs? What additional information are the readers getting from both plots? Likewise, additional text addressing why frequency domain instruments were used, and why time domain instruments were not used, would be helpful.

2. In figure 3 it is challenging to see the conformation change in the structures from the diagrams. It is quite difficult to see the blue and green dotted lines in the structures. It would be helpful to either simplify these diagrams, increase the weight of the lines, or add distances to these plots.

3. In figure 3-4 the authors do not present the calculated distances from the steady-state donor-quenching FRET changes. It would be helpful to see how well the distances generated from steady-state decreases in fluorescence match the expected backbone or modeled distances.

4. Did the authors try other metals with Acd (nickel, cobalt, etc.)?

5. The plots in Figure 6 are difficult to follow with the current color/shape scheme. The diamonds and circles are hard to distinguish. Please change these plots to make them easier to follow. I would suggest single open and closed circles (sizes could be adjusted) with different colors.

6. It would be helpful to plot MBP-295Acd-HH and MBP-295Acd-C (figure 6 supp) on the same graph to emphasize the radical change in probability distances as a function of the ligand and acceptor. Likewise, please add additional discussion on the difference in absolute distances (r) between apo and holo shown in this plot between the two acceptor systems. These are useful ideas for the field.

7. Can the authors comment on the fact the holo (pink) distances in 7D are not on the "universal circle" but still contain multi-exponential features?

---

## [Author Response]

Essential Revisions:The reviewers judge that the use of the non-canonical amino acid (NCA) Acd is an significant advance over ANAP and other fluorescent NCAs. The authors present a strong case for these advantages using the purified MBP as a protein model. However, the reviewers argue that MBP is a rare case of a well characterized and behaved soluble protein and that for ACd and the methods used to really be an improvement over available approaches, the authors should demonstrate its use in a membrane protein or at least a membrane-based system.

Our goal here was to establish the method with a model protein with characterized structures. While our future goals are to apply this technique to other proteins, including membrane proteins, this cannot be done in a rigorous manner within the scope of this manuscript. However, we do wish to point out that we have recently published new work in which we incorporate Acd into both cytoplasmic proteins (GFP, α-synuclein, and calmodulin, as well as MBP) and membrane proteins (HCN channel, insulin receptor) and study them in living mammalian cells (Jones et al., Chemical Science, In Press). We show that the long fluorescence lifetime of Acd can be used to distinguish the proteins of interest from cellular autofluorescence and to localize proteins in the correct subcellular compartments. We have also added a section to the Discussion highlighting that cellular autofluorescence likely limits use of our tmFRET approach with Acd to in vitro systems. The new manuscript is available in bioRxiv and has now been included as a companion paper to this submission.

Reviewer #1:The authors claimed that for experiments using Cu2+-TETAC as an acceptor, 8% contribution was from donor-only. However, how such a fraction of donor-only was estimated is missing in the text. The authors should clarify. Furthermore, for the experiments using di-histidine as an acceptor, the authors didn't provide enough information to show that the donor-only population was negligible. In Figure 7D, the center of mass of the measured data for 10 mM maltose (magenta) on the phasor plot seems to shift a bit towards the direction of donor-only population compared with the center of f_model, which may indicate the existence of donor-only population. If the donor-only fraction is not negligible, then the tmFRET results in Figure 7 should be fitted with 3-component (donor-only, Apo, and holo), rather than 2-component model.

The data supporting use of an 8% contribution from donor-only in Cu^2+^-TETAC experiments are shown in Figure 3 —figure supplement 2 and referenced in the main text and methods. For experiments using Cu^2+^ binding to di-histidine, we did not include a donor-only population in our model because our previously published work using Anap as a donor with Cu^2+^ bound to this same di-histidine showed a K_D_ of 3 µM (Gordon, et al., eLife, 2018). Unlike cysteines, which may oxidize and become unable to react with TETAC, di-histidine is unlikely to be refractory to binding Cu^2+^ at over 3,000 times its K_D_. We did model the effect of assuming 5%, 10%, and 20% unlabeled acceptor on the fits to our frequency domain lifetime data but doing so did not improve the fit to the 10 mM maltose data (in fact, doing so made the fits worse). Although we cannot rule out that some protein is not labeled with acceptor, we have no experimentally justified reason for adding an additional free parameter.

The donor is still a dipole, and the Foerster equation only holds when sufficient rotational mobility exists. This is why FRET usually uses the long linkers, which they stress as a disadvantage. Maybe the authors can comments on this. I believe their long linker statement might only apply to their acceptor, but for their donor it is actually a concern. Polarization effects would thus need to be discussed and accounted for.

The reviewer seems to be asking whether the rotational mobility of the donor and acceptor justifies our assumption that ĸ^2^ = 2/3. There are at least three reasons why we think this assumption is justified for tmFRET, which are now included in the manuscript text:

a) The metal ion acceptor has multiple transition dipole moments and even with limited mobility should sample orientational space sufficiently to be nearly freely rotating. In the case of a mixed polarization donor-acceptor pair, in which one is freely rotating, and one is immobile, 1/3 < ĸ^2^ < 4/3. Thus, even if our donor were immobile, an assumption of ĸ^2^ = 2/3 gives a maximum error in R_0_ of about ±11% (Haas, et al., Biochemistry, 1978).

b) Our data reveal a distribution of donor-acceptor distances, introducing additional sources of randomized orientation and further reducing the potential error due to the ĸ^2^ = 2/3 assumption.

c) Using ĸ^2^ = 2/3 for our R_0_ calculations in this and other tmFRET studies (e.g. Taraska et al., Nature Methods, 2009; Yu et al., Structure, 2013; Gordon et al., eLife, 2018) gives the expected distances based on known structures.

The authors claimed that Anap, but not Acd, appeared to undergo a site-specific chemical change in cells. Yet, a small but noticeable shift can be seen for MBP-295Acd in in Figure 1 supp. 1A. Can the authors give some comments on that? Given the results that mutation of Y307 tyrosine to F can prevent the quenching of MBP-295Acd, one may attribute the shift of MBP-295Anap or MPB-295Acd in Figure 1 supp 1A,B to the slow energy transferring or blue-shifting by Y307 during cell culture. Since Acd is quenched more slowly than Anap, the shift may show not as much as Anap. In such sense, is it better to harvest cells 24hr after transfection rather than 48hr if one wants to use Acd to measure other kinds of proteins without site-mutations?

The experiments in Figure 1 —figure supplement 1 were carried out under denaturing conditions so that Y307 is no longer quenching the donor fluorescence. We do not see any effect of time in culture on Acd fluorescence. The very small difference between the spectra shown in the figure is due to imperfect subtraction of the background from the small signal at 24 hours.

The authors mentioned that R0 value calculated for the Acd/Cu2+-TETAC FRET pair (14.9 Á…) is lower than for the Anap/Cu2+-TETAC FRET pair (17.2 Á…). Did the authors measure the R0 for all the mutants, e.g. MBP-295-C, MBP-295-307F, and MBP-322-C. Do they show same R0 or not for same noncanonical amino acid? What is the R0 for Acd/Cu2+-di-histidine FRET pair? The author should provide a list for that.

We have added a new table (Table 2) with the R_0_ values used in the manuscript. We also include a new figure, Figure 8 —figure supplement 1, showing the emission spectra of the Acd-incorporating proteins. The absorption spectra for Cu^2+^-TETAC and Cu^2+^-di-histidine are also shown. This figure demonstrates that the spectra are indistinguishable from each other, and from free Acd, at wavelengths above 490 nm where the overlap with the acceptors occurs. The text associated with this new figure explains that we used the same R_0_ value for all Acd/Cu^2+^-TETAC pairs because we cannot justify using different values based on the spectra.

Reviewer #2:While the authors present their results as a way to obtain dynamic information; it should be pointed out that as it stands the method can estimate equilibrium constants but not rate constants, which are needed to say that one has access to the dynamics.

Although we don’t measure rates here, tmFRET can be used to measure rates. We have made sure we use term “dynamics” in the hypothetical sense and don’t make the claim to measure dynamics in this work.

– I think the presentation will benefit from a figure showing the raw fluorescence data and explaining how the phase and amplitude are obtained and interpreted.

We appreciate this suggestion, which is echoed by reviewer #3 as well. In response we have added a new figure (Figure 8 —figure supplement 1), and associated text in the methods section, to walk the reader through time domain and frequency domain measurements. We illustrate donor-only and donor-acceptor FRET fluorescence lifetimes and better explain how the phase delay and modulation ratio change as a result of FRET.

– While the authors suggest a general applicability of the method, they have only show its application to a small, well behaved soluble protein. Have the authors applied it, even if preliminarily, to membrane proteins or other, more complex proteins?

As noted in our response to the “Essential Revisions,” our goal here was to establish the method with a model protein with characterized structures. While our future goals are to apply this technique to other proteins, including membrane proteins, this cannot be done in a rigorous manner within the scope of this manuscript. However, we do wish to point out that we have recently published new work in which we incorporate Acd into both cytoplasmic proteins (GFP, α-synuclein, and calmodulin, as well as MBP) and membrane proteins (HCN channel, insulin receptor) and study them in living mammalian cells (Jones et al., Chemical Science, In Press). We show that the long fluorescence lifetime of Acd can be used to distinguish the proteins of interest from cellular autofluorescence and to localize proteins in the correct subcellular compartments. We have also added a section to the Discussion highlighting that cellular autofluorescence likely limits use of our tmFRET approach with Acd to in vitro systems. The new manuscript is available in bioRxiv and has now been included as a companion paper to this submission.

– The distribution of distances is assumed to be Gaussian, please provide a justification for this assumption. Is it a necessary part of the method? What happens when the distribution of distances is skewed?

A Gaussian distance distribution need not be assumed, but the model requires some form of the distance distribution to be specified. Similar assumptions have been made in analysis of DEER spectroscopy, where the high-resolution distance distributions are well-fit by Gaussians or sums of Gaussians. We have added a discussion of the form of the distribution to indicate that it need not be Gaussian.

– While Acd is presented as an improvement over ANAP, the differences in photo physic properties really not that great, except the lifetime. Did you try to carry out similar lifetime measurements with ANAP? It seems the resolution of the instrumentation should be sufficient.

Acd has a longer lifetime, is single exponential, and bleaches much more slowly than Anap. The longer lifetime provides a greater dynamic range for measuring FRET, and the single-exponential lifetime requires fewer assumptions in the analysis. In preliminary experiments with Anap, the resolution of our instrument was more than adequate, but the multi-exponential decay of Anap was problematic for the analysis. The photostabilities of Acd and Anap (as well as other fluorescent amino acids) are characterized in the forthcoming Chemical Science manuscript included for review.

Reviewer #3:1. Understanding frequency domain plots can be tricky for the non-specialist. Standard lifetime plots in the time domain are generally (a bit) more intuitive and common. The authors do a good job of addressing the plots of lifetime in the frequency domain but I would suggest a more robust description of these plots and their expected changes in the main text to aid the reader in understanding these data. This is particularly important because this method will be generally useful for a broad audience. For example, why are both the modulation ration and phase delay necessary to plot on the graphs? What additional information are the readers getting from both plots? Likewise, additional text addressing why frequency domain instruments were used, and why time domain instruments were not used, would be helpful.

At the suggestion of this reviewer and reviewer #2, we have added a new figure (Figure 8 – figure supplement 1), and associated text in the methods section, to walk the reader through time domain and frequency domain measurements. We illustrate donor-only and donor-acceptor FRET fluorescence lifetimes and better explain how the phase delay and modulation ratio change as a result of FRET.

2. In figure 3 it is challenging to see the conformation change in the structures from the diagrams. It is quite difficult to see the blue and green dotted lines in the structures. It would be helpful to either simplify these diagrams, increase the weight of the lines, or add distances to these plots.

We have simplified the figure by removing the lines. We have also added a table (Table 2) that lists all the relevant distances.

3. In figure 3-4 the authors do not present the calculated distances from the steady-state donor-quenching FRET changes. It would be helpful to see how well the distances generated from steady-state decreases in fluorescence match the expected backbone or modeled distances.

We have added a new figure (Figure 4 —figure supplement 1) that directly compares the predicted distances to those calculated from steady-state and time-resolved FRET measurements. As we previously showed (Gordon, eLife, 2018) the distances calculated from steady-state FRET underpredict the maltose-dependent distance changes, an effect partly remedied using fluorescence lifetimes.

4. Did the authors try other metals with Acd (nickel, cobalt, etc.)?

We did not, though we have also used both Ni^2+^ and Co^2+^ in our previous publication with Anap (Gordon et al., 2018).

5. The plots in Figure 6 are difficult to follow with the current color/shape scheme. The diamonds and circles are hard to distinguish. Please change these plots to make them easier to follow. I would suggest single open and closed circles (sizes could be adjusted) with different colors.

We have taken the reviewer’s suggestion and altered the size and shape of the data points in Figure 5c, 6ad, and 6 sup 1 to make it easier to distinguish between the different markers.

6. It would be helpful to plot MBP-295Acd-HH and MBP-295Acd-C (figure 6 supp) on the same graph to emphasize the radical change in probability distances as a function of the ligand and acceptor. Likewise, please add additional discussion on the difference in absolute distances (r) between apo and holo shown in this plot between the two acceptor systems. These are useful ideas for the field.

As suggested by the reviewer, we added the fits from Figure 6 —figure supplement 2A onto Figure 6 – figure supplement 2C and modified figure legend to facilitate direct comparison of the predicted donor-acceptor distances using the different methods of labeling with Cu^2+^. We have also added a new figure, Figure 8 —figure supplement 1, and associated text, which highlights the differences in distances and maltose-dependent distance changes for Cu^2+^-TETAC compared to Cu^2+^-di-histidine.

7. Can the authors comment on the fact the holo (pink) distances in 7D are not on the "universal circle" but still contain multi-exponential features?

The holo peak is expected to be off circle because it represents a distribution of distances and is therefore non exponential. This highlights that even a single functional state can be heterogeneous.